# PARAMETRIC COPULA-GP MODEL FOR ANALYZING MULTIDIMENSIONAL NEURONAL AND BEHAVIORAL RELATIONSHIPS

## ABSTRACT

One of the main challenges in current systems neuroscience is the analysis of high-dimensional neuronal and behavioral data that are characterized by different statistics and timescales of the recorded variables. We propose a parametric copula model which separates the statistics of the individual variables from their dependence structure, and escapes the curse of dimensionality by using vine copula constructions. We use a Bayesian framework with Gaussian Process (GP) priors over copula parameters, conditioned on a continuous task-related variable. We improve the flexibility of this method by 1) using non-parametric conditional (rather than unconditional) marginals; 2) linearly mixing copula elements with qualitatively different tail dependencies. We validate the model on synthetic data and compare its performance in estimating mutual information against the commonly used non-parametric algorithms. Our model provides accurate information estimates when the dependencies in the data match the parametric copulas used in our framework. Moreover, when the exact density estimation with a parametric model is not possible, our Copula-GP model is still able to provide reasonable information estimates, close to the ground truth and comparable to those obtained with a neural network estimator. Finally, we apply our framework to real neuronal and behavioral recordings obtained in awake mice. We demonstrate the ability of our framework to 1) produce accurate and interpretable bivariate models for the analysis of inter-neuronal noise correlations or behavioral modulations; 2) expand to more than 100 dimensions and measure information content in the whole-population statistics. These results demonstrate that the Copula-GP framework is particularly useful for the analysis of complex multidimensional relationships between neuronal, sensory and behavioral data.

## 1 INTRODUCTION

Recent advances in imaging and recording techniques have enabled monitoring the activity of hundreds to several thousands of neurons simultaneously (Jun et al., 2017; Helmchen, 2009; Dombeck et al., 2007). These recordings can be made in awake animals engaged in specifically designed tasks or natural behavior (Stringer et al., 2019; Pakan et al., 2018a;b), which further augments these already large datasets with a variety of behavioral variables. These complex high dimensional datasets necessitate the development of novel analytical approaches (Saxena & Cunningham, 2019; Stevenson & Kording, 2011; Staude et al., 2010) to address two central questions of systems and behavioral neuroscience: how do populations of neurons encode information? And how does this neuronal activity correspond to the observed behavior? In machine learning terms, both of these questions translate into understanding the high-dimensional multivariate dependencies between the recorded variables (Kohn et al., 2016; Shimazaki et al., 2012; Ince et al., 2010; Shamir & Sompolinsky, 2004).

There are two major methods suitable for recording the activity of large populations of neurons from behaving animals: the multi-electrode probes (Jun et al., 2017), and calcium imaging methods (Grienberger et al., 2015; Helmchen, 2009; Dombeck et al., 2007) that use changes in intracellular calcium concentration as a proxy for neuronal spiking activity at a lower temporal precision. While neuronal spiking occurs on a temporal scale of milliseconds, the behavior spans the timescales from milliseconds to hours and even days (Mathis et al., 2018). As a result, the recorded neuronal

and behavioral variables may operate at different timescales and exhibit different statistics, which further complicates the statistical analysis of these datasets.

The natural approach to modeling statistical dependencies between the variables with drastically different statistics is based on *copulas*, which separate marginal (i.e. single variable) statistics from the dependence structure (Joe, 2014). For this reason, copula models are particularly effective for *mutual information* estimation (Jenison & Reale, 2004; Calsaverini & Vicente, 2009b), which quantifies how much knowing one variable reduces the uncertainty about another variable (Quiroga & Panzeri, 2009). Copula models can also escape the 'curse of dimensionality' by factorising the multi-dimensional dependence into pair-copula constructions called *vines* (Aas et al., 2009; Czado, 2010).

Copula models have been successfully applied to spiking activity (Onken et al., 2009; Hu et al., 2015; Shahbaba et al., 2014; Berkes et al., 2009), 2-photon calcium recordings (Safaai, 2019) and multi-modal neuronal datasets (Onken & Panzeri, 2016). However, these models assumed that the dependence between variables was static, whereas in neuronal recordings it may be dynamic or modulated by behavioral context (Doiron et al., 2016; Shimazaki et al., 2012). Therefore, it might be helpful to explicitly model the continuous time- or context-dependent changes in the relationships between variables, which reflect changes in an underlying computation.

Here, we extend a copula-based approach by adding explicit conditional dependence to the parameters of the copula model, approximating these latent dependencies with Gaussian Processes (GP). It was previously shown that such a combination of parametric copula models with GP priors outperforms static copula models (Lopez-Paz et al., 2013) and even dynamic copula models on many real-world datasets, including weather forecasts, geological data or stock market data (Hernández-Lobato et al., 2013). Yet, this method has never been applied to neuronal recordings before.

In this work, we increase the complexity of both marginal and copula models in order to adequately describe the complex dependencies commonly observed in neuronal data. In particular, we use conditional marginal models to account for changes of the single neuron statistics and mixtures of parametric copula models to account for changes in tail dependencies. We also improve the scalability of the method by using stochastic variational inference. We develop model selection algorithms, based on the fully-Bayesian Watanabe–Akaike information criterion (WAIC). Finally and most importantly, we demonstrate that our model is suitable for estimating mutual information. It performs especially well when the parametric model can closely approximate the target distribution. When it is not the case, our copula mixture model demonstrates sufficient flexibility and provides close information estimates, comparable to the best state-of-the-art non-parametric information estimators.

The goal of this paper is to propose and validate the statistical Copula-GP method, and illustrate that it combines multiple desirable properties for neuroscience applications: interpretability of parametric copula models, accuracy in density and information estimation and scalability to large datasets. We first introduce the copula mixture models and propose model selection algorithms (Sec. 2). We then validate our model on synthetic data and compare its performance against other commonly used information estimators (Sec. 3). Next, we demonstrate the utility of the method on real neuronal and behavioral data (Sec. 4). We show that our Copula-GP method can produce bivariate models that emphasize the qualitative changes in tail dependencies and estimate mutual information that exposes the structure of the task without providing any explicit cues to the model. Finally, we measure information content in the whole dataset with 5 behavioral variables and more than 100 neurons.

## 2    PARAMETRIC COPULA MIXTURES WITH GAUSSIAN PROCESS PRIORS

Our model is based on copulas: multivariate distributions with uniform marginals. Sklar's theorem (Sklar, 1959) states that any multivariate joint distribution can be written in terms of univariate marginal distribution functions $p(y_i)$ and a unique copula which characterizes the dependence structure: $p(y_1, \ldots, y_N) = c(F_1(y_1) \ldots F_N(y_N)) \times \prod_{i=1}^{N} p(y_i)$. Here, $F_i(\cdot)$ are the marginal cumulative distribution functions (CDF). Thus, for each $i$, $F_i(y_i)$ is uniformly distributed on [0,1].

For high dimensional datasets ($\dim \mathbf{y}$), maximum likelihood estimation for copula parameters may become computationally challenging. The two-stage inference for margins (IFM) training scheme is typically used in this case (Joe, 2005). First, univariate marginals are estimated and used to map the data onto a multidimensional unit cube. Second, the parameters of the copula model are inferred.

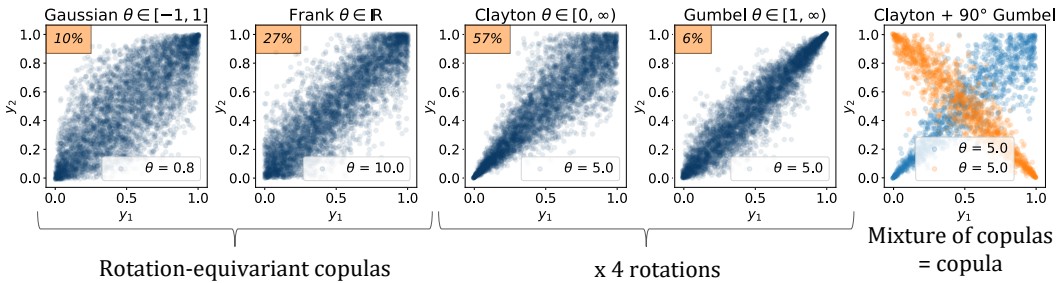

Figure 1: Copula families used in the mixture models in our framework. The percentage in the upper-left corner shows how often each of the families was selected to be used in a copula mixture for pairwise relationships in the real neuronal data from Pakan et al. (2018a) (see Sec. 4).

**Conditional copulas** Following the approach by Hernández-Lobato et al. (2013), we are using Gaussian Processes (GP) to model the conditional dependencies of copula parameters:

$$p(\mathbf{y}|x) = c\Big(F_1(y_1|x), \dots, F_N(y_N|x)\Big|x\Big) \times \left[\prod_{i=1}^{N} p(y_i|x)\right]. \tag{1}$$

In the most general case, the marginal PDFs $p(y_i|x)$ and CDFs $F_i(y_i|x)$ and the copula $c(\dots|x)$ itself can all be conditioned on $x$. In our framework, $x$ is assumed to be one-dimensional. A Gaussian Process is ideally suited for copula parametrization, as it provides an estimate of the uncertainty in model parameters, which we utilize in our model selection process (Sec. 2.1).

**Conditional marginals** Previous works on conditional copulas (Hernández-Lobato et al., 2013) relied on the assumption that marginal distributions remain constant. This assumption might not hold for some real-life datasets, including neuronal recordings. Thus, we propose to use **conditional** marginal distributions instead. Note, that the interpretation of the copula model itself would depend on the assumptions made for the marginal distributions (see Appx. C.1 for further discussion).

In order to estimate marginal CDFs $F(y_i|x)$, we use the non-parametric fastKDE (O'Brien et al., 2016) algorithm, which allows for direct estimation of the conditional distributions. These CDFs are then used to map the data onto a unit hypercube using the probability integral transform: $F(y_i|x) \to u_i \sim u_{[0,1]}$, such that $u_i$ is uniformly distributed for any $x$.

**Bivariate copula families** We use 4 copula families as the building blocks for our copula models: Gaussian, Frank, Clayton and Gumbel copulas (Fig. 1). All of these families have a single parameter, corresponding to the rank correlation (Table 1). We also use rotated variants (90°, 180°, 270°) of Clayton and Gumbel copulas in order to express upper tail dependencies and negative correlation.

Table 1: Bivariate copula families and their GPLink functions

| Copula | Domain | $\mathrm{GPLink}(f): \mathbb{R} \to \mathrm{dom}(c_j)$ |
| --- | --- | --- |
| Independence | – | – |
| Gaussian | [-1,1] | $\mathrm{Erf}(f/1.4)$ |
| Frank | $(-\infty, \infty)$ | $0.3 \cdot f + \mathrm{sign}(f) \cdot (0.3 \cdot f)^2$ |
| Clayton | $[0,\infty)$ | $\mathrm{Exp}(0.3 \cdot f)$ |
| Gumbel | $[1,\infty)$ | $1 + \mathrm{Exp}(0.3 \cdot f)$ |

Since we are primarily focused on the analysis of neuronal data, we have first visualized the dependencies in calcium signal recordings after a probability integral transform, yielding empirical conditional copulas. As a distinct feature in neuronal datasets, we observed changes in tail dependencies with regard to the conditioning variable. Since none of the aforementioned families alone could describe such conditional dependency, we combined multiple copulas into a linear *mixture model*: $c(\mathbf{u}|x) = \sum_{j=1}^{M} \phi_j(x) c_j(\mathbf{u}; \theta_j(x))$, where $M$ is the number of elements, $\phi_j(x)$ is the concentration of the $j$th copula in a mixture, $c_j$ is the pdf of the $j$th copula, and $\theta_j$ is its parameter.

Each of the copula families includes the Independence copula as a special case. To resolve this over-completeness, we add the Independence copula as a separate model with zero parameters (Table 1). For independent variables $\mathbf{y}_{ind}$, the Independence model will be preferred over the other models in our model selection algorithm (Sec. 2.1), since it has the smallest number of parameters.

**Gaussian Process priors**    We parametrize each copula in the mixture model with an independent latent GP: $\mathbf{f} \sim \mathcal{N}(\mu \times \mathbf{1}, K_\lambda(x, x))$. For each copula family, we constructed GPLink functions (Table 1) that map the GP variable onto the copula parameter domain: $\theta_j = \text{GPlink}_{c_j}(f_j), \mathbb{R} \rightarrow \text{dom}(c_j)$. Next, we also use GP to parametrize concentrations $\phi_j(x)$, which are defined on a simplex ($\sum \phi = 1$):

$$\phi_j = (1 - t_j) \prod_{m=1}^{j-1} t_m, \quad t_m = \Phi\left(\widetilde{f}_m + \Phi^{-1}\left(\frac{M - m - 1}{M - m}\right)\right), \quad t_M = 0,$$

where $\Phi$ is a CDF of a standard normal distribution and $\widetilde{\mathbf{f}}_\mathbf{m} \sim \mathcal{N}(\widetilde{\mu}_m \times \mathbf{1}, \widetilde{K}_{\widetilde{\lambda}_m}(x, x))$. We use the RBF kernel $K_\lambda(x, x)$ with bandwidth parameter $\lambda$. Therefore, the whole mixture model with $M$ copula elements is parameterized by $[2M - 1]$ independent GPs and requires $[2M - 1]$ hyperparameters: $\{\lambda\}_M$ for $\boldsymbol{\theta}$ and $\{\widetilde{\lambda}\}_{M-1}$ for $\boldsymbol{\phi}$.

**Approximate Inference**    Since our model has latent variables with GP priors and intractable posterior distribution, direct maximum likelihood Type-II estimation is not possible and an approximate inference is needed. Such an inference problem with copula models has previously been solved with the expectation propagation algorithm (Hernández-Lobato et al. (2013), see comparison in Appx. B.4), which was not suitable for large scale data. Recently, a number of scalable approximate inference methods were developed, including stochastic variational inference (SVI) (Titsias, 2009; Cheng & Boots, 2017), scalable expectation propagation (SEP) (Hernández-Lobato & Hernández-Lobato, 2016), and MCMC based algorithms (Hensman et al., 2015), as well as a scalable exact GP (Wang et al., 2019). We chose to use SVI due to availability of the well-established GPU-accelerated libraries: PyTorch (Paszke et al., 2017) and GPyTorch (Gardner et al., 2018).

## 2.1 BAYESIAN MODEL SELECTION

We use the Watanabe–Akaike information criterion (WAIC, Watanabe (2013)) for model selection. WAIC is a fully Bayesian approach to estimating the Akaike information criterion (AIC) (see Eq. 31 in the original paper by Watanabe (2013)). The main advantage of the method is that it avoids the empirical estimation of the effective number of parameters, which is often used for approximation of the out-of-sample bias. It starts with the estimation of the log pointwise posterior predictive density (lppd) (Gelman et al., 2014):

$$\widehat{\text{lppd}} = \sum_{i=1}^{N} \log\left(\frac{1}{S}\sum_{s=1}^{S} p(y_i|\theta^s)\right), \qquad p_{\text{WAIC}} = \sum_{i=1}^{N} V_{s=1}^{S}\Big(\log p(y_i|\theta^s)\Big),$$

where $\{\theta^s\}_S$ is a draw from a posterior distribution, which must be large enough to represent the posterior. Next, the $p_{\text{WAIC}}$ approximates the bias correction, where $V_{s=1}^{S}$ represents sample variance. Therefore, the bias-corrected estimate of the log pointwise posterior predictive density is given by: $\widehat{\text{elppd}}_{\text{WAIC}} = \text{lppd} - p_{\text{WAIC}} = -N \cdot \text{WAIC}_{original}$.

In the model selection process, we aim to choose the model with the lowest WAIC. Since our copula probability densities are continuous, their values can exceed 1 and the resulting WAIC is typically negative. Zero WAIC corresponds to the Independence model ($\text{pdf} = 1$ on the whole unit square).

Since the total number of combinations of 10 copula elements (Fig. 1, considering rotations) is large, exhaustive search for the optimal model is not feasible. In our framework, we propose two model algorithms for constructing close-to-optimal copula mixtures: *greedy* and *heuristic* (see Appx. A.4 for details). The greedy algorithm is universal and can be used with any other copula families without adjustment, while the heuristic algorithm is fine-tuned to the specific copula families used in this paper (Fig. 1). Both model selection algorithms were able to select the correct 1- and 2-component model on simulated data and at least find a close approximation (within $\text{WAIC}_{tol} = 0.005$) for more complex models (see validation of model selection in Appx. B).

## 2.2 Entropy and mutual information

Our framework provides tools for efficient sampling from the conditional distribution and for calculating the probability density $p(\mathbf{y}|x)$. Therefore, for each $x{=}t$ the entropy $H(\mathbf{y}|x{=}t)$ can be estimated using Monte Carlo (MC) integration: $H(\mathbf{y}|x{=}t) = -\mathbb{E}_{p(\mathbf{y}|x{=}t)} \log p(\mathbf{y}|x{=}t)$. The probability $p(\mathbf{y}|x{=}t)$ factorizes into the conditional copula density and marginal densities (1), hence the entropy also factorizes (Jenison & Reale, 2004) as $H(\mathbf{y}|x{=}t) = \sum H(y_i|x{=}t) + H_c(\mathbf{u}^x|x{=}t)$, where $\mathbf{u}^x = \mathbf{F}(\mathbf{y}|x)$. The conditional entropy can be integrated as $H(\mathbf{y}|x) = \sum_{i=1}^{N} H(y_i|x) + \int H_c(\mathbf{u}^x|x{=}t)p(t)dt$, separating the entropy of the marginals $\{y_i\}_N$ from the copula entropy.

Now, $I(x, \mathbf{y}) = I(x, \mathbf{G}(\mathbf{y}))$ if $\mathbf{G}(\mathbf{y})$ is 1) a homeomorphism, 2) independent of $x$ (Kraskov et al., 2004). If marginal statistics are independent of $x$, then the probability integral transform $\mathbf{u} = \mathbf{F}(\mathbf{y})$ satisfies both requirements, and $I(x, \mathbf{y}) = I(x, \mathbf{u})$. Then, in order to calculate the mutual information $I(x, \mathbf{u}) := H(\mathbf{u}) - H(\mathbf{u}|x)$, we must also rewrite it using only the conditional distribution $p(\mathbf{u}|x)$, which is modelled with our Copula-GP model. This can be done as follows:

$$I(x, \mathbf{u}) = H(\mathbf{u}) - \int H(\mathbf{u}|x = t)p(t)dt = \mathbb{E}_{p(\mathbf{u},x)} \log p(\mathbf{u}|x) - \mathbb{E}_{p(\mathbf{u})} \log \mathbb{E}_{p(x)} p(\mathbf{u}|x). \quad (2)$$

The last term in (2) involves nested integration, which is computationally difficult and does not scale well with $N = \dim \mathbf{u}$. Therefore, we propose an alternative way of estimating $I(x, \mathbf{y})$, which avoids double integration and allows us to use the marginals conditioned on $x$ ($\mathbf{u}^x = \mathbf{F}(\mathbf{y}|x)$), providing a better estimate of $H(\mathbf{y}|x)$. We can use two separate copula models, one for estimating $p(\mathbf{y})$ and calculating $H(\mathbf{y})$, and another one for estimating $p(\mathbf{y}|x)$ and calculating $H(\mathbf{y}|x)$:

$$I(x, \mathbf{y}) = \sum_{i=1}^{N} I(x, y_i) + H_c(u_1, \ldots, u_N) - \int H_c(u_1^x, \ldots, u_N^x|s = t)p(t)dt, \quad (3)$$

where both entropy terms are estimated with MC integration. Here we only integrate over the unit cube $[0, 1]^N$ and then $\dom x$, whereas (2) required integration over $[0, 1]^N \times \dom x$.

The performance of both (2) and (3) critically depends on the approximation of the dependence structure, i.e. how well the parametric copula approximates the true copula probability density. If the joint distribution $p(y_1 \ldots y_N)$ has a complex dependence structure, as we will see in synthetic examples, then the mixture of parametric copulas may provide a poor approximation of $p(\mathbf{y})$ and overestimate $H_c(u_1, \ldots, u_N)$, thereby overestimating $I(x, \mathbf{y})$. The direct integration (2), on the other hand, typically underestimates the $I(x, \mathbf{y})$ due to imperfect approximation of $p(\mathbf{y}|x)$, and under assumption that the marginals can be considered independent of $x$.

We further refer to the direct integration approach (2) as "Copula-GP integrated" and to the alternative approach (3) as "Copula-GP estimated" and assess both of them on synthetic and real data.

## 2.3 Copula vine constructions

High-dimensional copulas can be constructed from bivariate copulas by organizing them into hierarchical structures called *copula vines* (Aas et al., 2009). In this paper, we focus on the *canonical vine* or *C-vine*, which factorizes the high-dimensional copula probability density function as follows:

$$c(\mathbf{u}) = \left[ \prod_{i=2}^{N} c_{1i}(u_1, u_i) \right] \times \left[ \prod_{i=2}^{N} \prod_{j=i+1}^{N} c_{ij|\{k\}_{k<i}} \left( F(u_i|\{u_k\}_{k<i}), F(u_j|\{u_k\}_{k<i}) \right) \right] \quad (4)$$

where $\{k\}_{k<i} = 1, \ldots, i - 1$ and $F(.|.)$ is a conditional CDF. Note, that all of the copulas in (4) can also be conditioned on $x$ via Copula-GP model. We choose the first variable $u_1$ to be the one with the highest rank correlation with the rest (sum of absolute values of pairwise Kendall's $\tau$), and condition all variables on the first one. We repeat the procedure until no variable is left (see Appx. A.5). It was shown by Czado et al. (2012) that this ordering facilitates C-vine modeling.

**Code availability** Code will be made available on GitHub upon paper acceptance.

# 3 VALIDATION ON ARTIFICIAL DATA

We compare our method with the other commonly used non-parametric algorithms for mutual information estimation: Kraskov-Stögbauer-Grassberger (KSG, Kraskov et al. (2004)), Bias-Improved-KSG by Gao et al. (BI-KSG, Gao et al. (2016)) and the Mutual Information Neural Estimator (MINE, Belghazi et al. (2018)). In this section, we use relatively low dimensional data ($\leq$10D), for which we can still directly calculate the true mutual information.

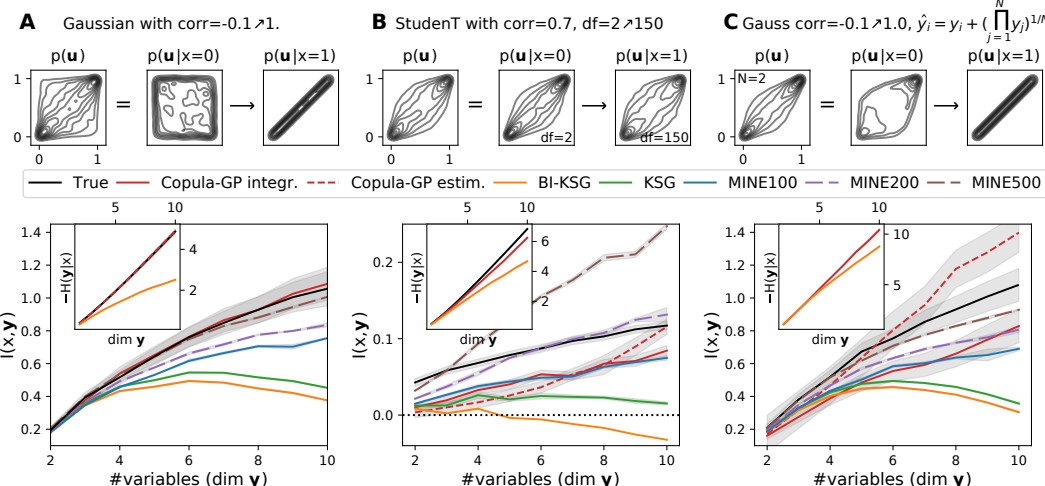

Figure 2: Conditional entropy $H(\mathbf{y}|x)$ and mutual information $I(x, \mathbf{y})$ measured by different methods on synthetic data. Upper row shows the dependency structures $p(\mathbf{u})$ and conditional dependency structures at the beginning and the end of the dom $x = [0, 1]$. **A** Multivariate Gaussian samples. **B** Multivariate Student T samples. **C** Multivariate Gaussian samples $\mathbf{y}$ (same as **A**), morphed into another distribution $p(\hat{\mathbf{y}})$ with a tail dependence, while $I(x, \mathbf{y}) = I(x, \hat{\mathbf{y}})$. Gray intervals show either standard error of mean (SE, 5 repetitions), or $\sqrt{(SE)^2 + (MC_{tol})^2}$ for integrated variables. Note, that MINE estimates depend on the choice of hyper-parameters (e.g. number of hidden units).

First, if there is no model mismatch, we expect our information estimates to be unbiased due to the MC estimator being unbiased (Luengo et al., 2020). We confirm this fact using the dataset sampled from a multivariate Gaussian distribution, with $cov(y_i, y_j) = \rho + (1 - \rho) \delta_{ij}$, where $\delta_{ij}$ is Kronecker's delta and $\rho = -0.1 + 1.1x, x \in [0, 1]$. Our algorithm selects a Gaussian copula on these data, which perfectly matches the true distribution. The same applies to any linear mixture of copulas from Tab. 1 as well (see Appx. B.3), and is covered by the automated tests on simulated data (see Appx. B.1). Therefore, Copula-GP measures both entropy and mutual information exactly (within integration tolerance, see Fig. 2A). The performance of the non-parametric methods on this dataset is lower. It was shown before that KSG and MINE both severely underestimate the MI for high-dimensional Gaussians with high correlation (e.g. see Fig. 1 in Belghazi et al. (2018)). The Copula-GP model (integrated) provides accurate estimates for highly correlated (up to $\rho = 0.999$, at least up to 20D) Gaussian distributions (see Appx. B.2).

Next, we test the Copula-GP performance on the Student T distribution, which can only be approximated by our copula mixtures, but would not exactly match any of the parametric copula families from Fig. 1. We keep the correlation coefficient $\rho$ fixed at $0.7$, and only change the number of degrees of freedom exponentially: $df = \exp(5x) + 1, x \in [0, 1]$. This makes the dataset particularly challenging, as all of the mutual information $I(x, \mathbf{y})$ is encoded in tail dependencies of $p(\mathbf{y}|x)$. The true $H(\mathbf{y}|x)$ of the Student T distribution was calculated analytically (see Eq. A.12 in Calsaverini & Vicente (2009a)) and $I(x, \mathbf{y})$ was integrated numerically according to (2) given the true $p(\mathbf{y}|x)$.

Fig. 2B shows that most of the methods underestimate $I(x, \mathbf{y})$. Copula-GP (integrated) and MINE (with 100 hidden units) provide the closest estimates. The training curve for MINE with more hidden units (200,500) showed signs of overfitting (abrupt changes in loss at certain permutations) and the resulting estimate was higher than the true $I(x, \mathbf{y})$ at higher dimensions. It was shown before that MINE provides inaccurate and inconsistent results on datasets with low $I(x, \mathbf{y})$ (Song & Ermon,

2019). We also demonstrate $I(x, \mathbf{y})$ estimation with a combination of two copula models for $H(\mathbf{y})$ and $H(\mathbf{y}|x)$: "Copula-GP estimated" (see Eq. 3). In lower dimensions, it captures less information than "Copula-GP integrated", but starts overestimating the true MI at higher dimensions, when the inaccuracy of the density estimation for $p(\mathbf{y})$ builds up. This shows the limitation of the "estimated" method, which can either underestimate or overestimate the correct value due to parametric model mismatch, whereas "integrated" method consistently underestimates the correct value. We conclude that Copula-GP and MINE demonstrate similar performance in this example, while KSG-based methods significantly underestimate $I(x, \mathbf{y})$ in higher dimensions.

Finally, we created another artificial dataset that is not related to any of the copula models used in our framework (Table 1). We achieved that by applying a homeomorphic transformation $\mathbf{F}(\mathbf{y})$ to a multivariate Gaussian distribution. Since the transformation is independent of the conditioning variable, it does not change the $I(x, \mathbf{y}) = I(x, \mathbf{F}(\mathbf{y}))$ (Kraskov et al., 2004). Therefore, we possess the true $I(x, \mathbf{y})$, which is the same as for the first example in Fig. 2A. Note, however, that there is no ground truth for the conditional entropy in this example, since $H(\mathbf{y}) \neq H(\mathbf{F}(\mathbf{y}))$. We transform the Gaussian copula samples $\mathbf{y} \in u_{[0,1]}^N$ from the first example as $\widetilde{y_i} = y_i + (\prod_{j=1}^N y_j)^{1/N}$ and again transform the marginals using the empirical probability integral transform $\mathbf{u} = \mathbf{F}(\widetilde{\mathbf{y}})$. Both conditional $p(\mathbf{u}|x)$ and unconditional $p(\mathbf{u})$ densities here do not match any of the parametric copulas from Table 1. As a result, "Copula-GP estimated" overestimated the correct value, while "Copula-GP integrated" underestimated it similarly to the MINE estimator with 100 hidden units.

Fig. 2 demonstrates that the performance of the parametric Copula-GP model critically depends on the match between the true probability density and the best mixture of parametric copula elements. When the parametric distribution matches the true distribution (e.g. Fig. 2A or Fig. A5), our Copula-GP framework provides unbiased estimates and predictably outperforms all non-parametric methods. Nonetheless, even when the exact reconstruction of the density is not possible (Figs. 2B-C), the mixtures of the copula models are still able to model the changes in tail dependencies, at least qualitatively. In those challenging examples, our method performs similarly to the neural-network based method (MINE) and still outperforms KSG-like methods.

## 4 VALIDATION ON REAL DATA

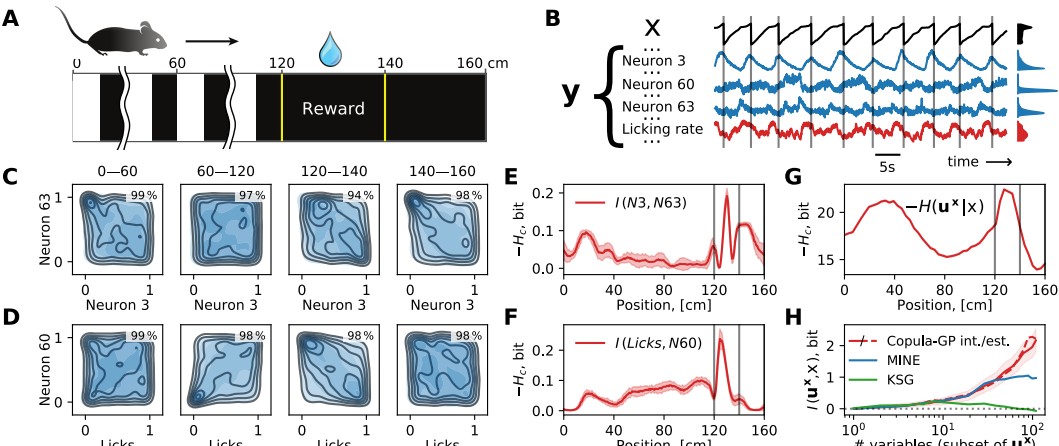

Figure 3: Applications of the Copula-GP framework to neuronal and behavioral data from the visual cortex of awake mice. **A** Schematic of the experimental task (Pakan et al., 2018a; Henschke et al., 2020) in virtual reality (VR); **B** Example traces from ten example trials: $x$ is a position in VR, $\mathbf{y}$ is a vector of neuronal recordings (blue) and behavioral variables (red); **C-D** Density plots for: the noise correlation (C) and the behavioral modulation (D) examples; **E-G** Conditional entropy for the bivariate examples (E-F) and the population-wide statistics (G); **H** Comparison of Copula-GP methods ("estimated" and "integrated") vs. non-parametric MI estimators on subsets of variables.

We investigate the dependencies observed in neuronal and behavioral data and showcase possible applications of the Copula-GP framework. We used two-photon calcium imaging data of neuronal

population activity in the primary visual cortex of mice engaged in a visuospatial navigation task in virtual reality (data from Henschke et al. (2020)). Briefly, the mice learned to run through a virtual corridor with vertical gratings on the walls (Fig. 3A, 0-120 cm) until they reached a reward zone (Fig. 3A, 120-140 cm), where they could get a reward by licking a reward spout. We condition our Copula-GP model on the position in the virtual environment $x$ and studied the joint distribution of the behavioral ($\tilde{y}_1 \ldots \tilde{y}_5$) and neuronal ($\tilde{y}_6 \ldots \tilde{y}_{109}$) variables ($\dim \mathbf{y}$=109). Fig. 3B shows a part of the dataset (trials 25-35 out of 130). The traces here demonstrate changes in the position $x$ of the mouse as well as the activity of 3 selected neurons and the licking rate. These variables have different patterns of activity depending on $x$ and different signal-to-noise ratios. Both differences are reflected in marginal statistics, which are shown on the right with the density plots of equal area.

**Constructing interpretable bivariate models**  We first studied bivariate relationships between neurons. In order to do this, we transformed the raw signals (shown in Fig. 3B) with a probability integral transform $\mathbf{u} = \mathbf{F}(\mathbf{y})$. We observed strong non-trivial changes in the dependence structure $c(\mathbf{u}|x)$ subject to the position in the virtual reality $x$ and related visual information (Fig. 3C). Such stimulus-related changes in the joint variability of two neuronal signals are commonly described as *noise correlations*. The Copula-GP model provides a more detailed description of the joint probability that goes beyond linear correlation analysis. In this example, the dependence structure is best characterized by a combination of Gaussian and Clayton copula (rotated by 90°). The density plots Fig. 3C demonstrate the match between the true density (outlines) and the copula model density (blue shades) for each part of the task. We measure the accuracy of the density estimation with the proportion of variance explained $R^2$, which shows how much of the variance of the variable $y_2$ can be predicted given the variable $y_1$ (see Eq. A1 in Appx. A.1). The average $\overline{R^2}$ for all $y_1$ is provided in the upper right corner of the density plots.

Next, we show that our model can be applied not only to the neuronal data, but also to any of the behavioral variables. Fig. 3D shows the dependence structure between one of the neurons and the licking rate. The best selected mixture model here is Frank + Clayton 0° + Gumbel 270°, which again provides an accurate estimate of the conditional dependence between the variables. Such copula mixture models allow us to analyze the contribution of the tail dependencies to the mutual information, and backtrack the outliers in order to relate them to certain trials or behaviors (see Appx. C.2). Therefore, Figs. 3C-D demonstrate that our Copula-GP model provides both an accurate fit for the probability distribution and an interpretable visualization of the dependence structure.

Figs. 3E-F show the absolute value of the conditional entropy $|H(\mathbf{u}^x|x)|$, which is equivalent to the mutual information between two variables $I(u_1^x, u_2^x)$. For both examples, the MI peaks in the reward zone. The bivariate Copula-GP models were agnostic of the reward mechanism in this task, yet they revealed the position of the reward zone as an anomaly in the mutual information.

**Measuring information content in a large neuronal population**  Finally, we constructed a C-vine describing the distribution between all neuronal and behavioral variables ($\dim \mathbf{u}^x = 109$) and measured the conditional entropy $H(\mathbf{u}^x|x)$ for all variables in the dataset $\{u_1^x \ldots u_{109}^x\}$. The conditional entropy in Fig. 3G peaks in the reward zone (similarly to Figs. 3E-F) and also at the beginning of the trial, where the velocity of the animal varied the most on the trial-to-trial basis.

While constructing the C-vine, we ordered the variables according to their pairwise rank correlations (see Sec. 2.3). We considered subsets of the first $N$ variables and measured the MI with the position for each subset. We compared the performance of our Copula-GP method on these subsets of $\mathbf{u}^x$ vs. KSG and MINE. Fig. 3H shows that all 3 methods provide similar results on subsets of up to 10 variables, yet in higher dimensions both MINE and KSG show smaller $I(x, \{u_{i<N}^x\})$ compared to our Copula-GP method, which agrees with the results obtained on the synthetic data (Fig. 2). The true values of $I(x, \{u_{i<N}^x\})$ are unknown, yet we expect the integrated Copula-GP (solid line) to underestimate the true value due to parametric model mismatch. The Copula-GP "estimated" (dashed line) can either under- or over-estimate the information (see eq. 3), but here it almost perfectly matches the "integrated" result, which suggests that the model was able to tightly approximate both $p(\mathbf{u}^x|x)$ and $p(\mathbf{u}^x)$, and, as a result (eq. 3), $I(x, \{u_{i<N}^x\})$. These results demonstrate superior performance of our Copula-GP model on high-dimensional neuronal data.

## 5 DISCUSSION

We have developed a Copula-GP framework for modeling conditional multivariate joint distributions. The method is based on linear mixtures of parametric copulas, which provide flexibility for estimating complex dependencies but still have interpretable parameters. We approximate conditional dependencies of the model parameters with Gaussian Processes which allow us to implement a Bayesian model selection procedure. The selected models combine the accuracy in density estimation with the interpretability of parametric copula models. We demonstrated the performance of our framework in mutual information estimation on synthetically generated data. Despite the general limitations of the parametric models, our framework performs either significantly better (when the true dependence matches a parametric model), or at least as good as the state-of-the-art non-parametric information estimators. Unlike black-box machine-learning methods for entropy estimation, our Copula-GP model uses interpretable elements (pair copulas) and latent GP parameters, which can be isolated, visualized, and analyzed (Appx. C). The framework is also well suited for describing the dependencies observed in neuronal and behavioral data. We demonstrated that the model scales well at least up to 109 variables and 21k samples, while theoretically, the parameter inference scales as $\mathcal{O}(n \cdot m^2)$, where $n$ is a number of samples and $m$ is the (effective) number of variables (see Appx. A.6). Alternative popular **non-copula** methods that could be applied to our real data, include GPFA or GLMs on the deconvolved spikes. Contrary to these approaches, copula methods allow us to model the dependencies between elements with utterly different statistics (e.g. licks vs. velocity). In addition, Copula-GP explicitly represents the dependencies as a function of position, revealing insightful information about the task structure. To the best of our knowledge, there are currently no other methods with this combination of features. The possible applications of Copula-GP include, but are not limited to, studying the neuronal population statistics, noise correlations, behavioral or contextual modulations. In summary, we demonstrated that the Copula-GP approach can make stochastic relationships explicit and generate accurate models of dependencies between neuronal responses, sensory stimuli, and behavior. Future work will focus on implementing model selection for the vine structure and improving the scalability of the mutual information estimation algorithm.

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

## A  APPENDIX METHODS

### A.1  GOODNESS-OF-FIT

We measure the accuracy of the density estimation with the proportion of variance explained $R^2$. We compare the empirical conditional CDF $\text{ecdf}(u_2|u_1 = y)$ vs. estimated conditional CDF $\text{ccdf}(u_2|u_1 = y)$ and calculate:

$$R^2(y) = 1 - \sum_{u_2} \left( \frac{\text{ecdf}(u_2|u_1 = y) - \text{ccdf}(u_2|u_1 = y)}{\text{ecdf}(u_2|u_1 = y) - \overline{u_2}} \right)^2, \tag{A1}$$

where $R^2(y)$ quantifies the portion of the total variance of $u_2$ that our copula model can explain given $u_1 = y$, and $\overline{u_2} = \overline{F(y_2)} = 0.5$. The sum was calculated for $u_2 = 0.05n, n = 0 \dots 20$.

Next, we select all of the samples from a certain interval of the task ($x \in [x_1, x_2]$) matching one of those shown in Figure 3 in the paper. We split these samples $u_1 \in [0, 1]$ into 20 equally sized bins: $\{I_i\}_{20}$. For each bin $I_i$, we calculate (A1). We evaluate $\text{ccdf}(u_2|u_1 = y_i) \approx \text{ccdf}(u_2|u_1 \in I_i)$ using a copula model from the center of mass of the considered interval of $x$: $x_\mu = \text{mean}(x)$ for samples $x \in [x_1, x_2]$. We use the average measure:

$$\overline{R^2} = \mathop{\mathbb{E}}_{p(u_1 \in I_i)} R^2\big(\text{mean}(u_1 \in I_i)\big), \tag{A2}$$

to characterize the goodness of fit for a bivariate copula model. Since $u_1$ is uniformly distributed on $[0, 1]$, the probabilities for each bin $p(u_1 \in I_i)$ are equal to $1/20$, and the resulting measure $\overline{R^2}$ is just an average $R^2$ from all bins. The results were largely insensitive to the number of bins (e.g. 20 vs. 100).

### A.2  VARIATIONAL INFERENCE

Since our model has latent variables with GP priors and intractable posterior distribution, the direct maximum likelihood Type-II estimation is not possible and an approximate inference is needed. We used stochastic variational inference (SVI) with a single evidence lower bound (Hensman et al., 2015):

$$\mathcal{L}_{\text{ELBO}} = \sum_{i=1}^{N} \mathop{\mathbb{E}}_{q(f_i)} \big[ \log p(y_i|f_i) \big] - \text{KL}[q(\mathbf{u})||p(\mathbf{u})], \tag{A3}$$

implemented as `VariationalELBO` in GPyTorch (Gardner et al., 2018). Here $N$ is the number of data samples, $\mathbf{u}$ are the inducing points, $q(\mathbf{u})$ is the variational distribution and $q(\mathbf{f}) = \int p(\mathbf{f}|\mathbf{u})q(\mathbf{u})d\mathbf{u}$.

Following the Wilson and Nickisch (2015) approach (KISS-GP), we then constrain the inducing points to a regular grid, which applies a deterministic relationship between $\mathbf{f}$ and $\mathbf{u}$. As a result, we only need to infer the variational distribution $q(\mathbf{u})$, but not the positions of $\mathbf{u}$. The number of grid points is one of the model hyper-parameters: `grid_size`.

Equation A3 enables joint optimization of the GP hyper-parameters (constant mean $\mu$ and two kernel parameters: scale and bandwidth) and parameters of the variational distribution $\mathbf{q}$ (mean and covariance at the inducing points: $\mathbf{u} \sim \mathcal{N}(\mu_u \times \mathbf{1}, \Sigma_u)$ ) (Hensman et al., 2015). We have empirically discovered by studying the convergence on synthetic data, that the best results are achieved when the learning rate for the GP hyper-parameters (`base_lr`) is much greater than the learning rate for the variational distribution parameters (`var_lr`, see Table A1).

**Priors**  For both the neuronal and the synthetic data, we use a standard normal prior $p(\mathbf{u}) \sim \mathcal{N}(\mathbf{0}, I)$ for a variational distribution. Note, that the parametrization for mixture models was chosen such that the aforementioned choice of the variational distribution prior with zero mean corresponds to *a priori* equal mixing coefficients $\phi_j = 1/M$ for $j = 1 \dots M$. In our experiments with the simulated and real neuronal data, we observed that the GP hyper-parameter optimisation problem often had 2 minima (which is a common situation, see Figure 5.5 on page 116 in Williams and Rasmussen (2006)). One of those corresponds to a short kernel lengthscale ($\lambda$) and low noise ($\min_{\mathbf{f}} \sigma^2$), which we interpret as overfitting. To prevent overfitting, we used $\lambda \sim \mathcal{N}(0.5, 0.2)$ prior on RBF kernel lengthscale parameter that allows the optimizer to approach the minima from the region of higher $\lambda$, ending up in the minimum with a larger lengthscale.

**Optimization**  We use the Adam optimizer with two learning rates for GP hyper-parameters (`base_lr`) and variational distribution parameters (`var_lr`). We monitor the loss (averaged over 50 steps) and its changes in the last 50 steps: $\Delta$ `loss = mean(loss[-100:-50]) - mean(loss[-50:])`. If the change becomes smaller than `check_waic`, then we evaluate the model WAIC and check if it is lower than $-\text{WAIC}_{tol}$. If it is higher, we consider that either the variables are independent, or the model does not match the data. Either way, this indicates that further optimisation is counterproductive. If the $\text{WAIC} < -\text{WAIC}_{tol}$, we proceed with the optimisation until the change of loss in 50 steps $\Delta loss$ becomes smaller than `loss_tol` (see Table A1).

**Effective learning rates for different families**  The coefficients in the GPLink functions for different copula families are also a part of model hyper-parameters. The choice of these coefficients affects the gradients of the log probability function. Since GPLink functions are nonlinear, they affect the gradients in various parameter ranges to a different extent. This results in variable convergence rates depending on the true copula parameters.

To address the problem of setting up these hyper-parameters, we have created the tests on synthetic data with different copula parameters. Using these tests, we manually adjusted these hyper-parameters such that the GP parameter inference converged in around 1000-2000 iterations for every copula family and parameter range. We have also multiplied the GP values corresponding to the mixture coefficients by 0.5, to effectively slow down the learning of the mixture coefficients $\phi$ compared to the copula coefficients $\theta$, which also facilitates the convergence.

**Hyper-parameter selection**  The hyper-parameters of our model (Table A1) were manually tuned, often considering the trade off between model accuracy and evaluation time. A more systematic hyper-parameter search might yield improved results and better determine the limits of model accuracy.

Table A1: Hyper-parameters of the bivariate Copula-GP model

| Hyper-parameter | Value | Description |
|---|---|---|
| `base_lr` | $10^{-2}$ | Learning rate for GP parameters |
| `var_lr` | $10^{-3}$ | Learning rate for variational distribution |
| `grid_size` | 128 | Number of inducing points for KISS-GP |
| `waic_tol` | 0.005 | Tolerance for WAIC estimation |
| `loss_tol` | $10^{-4}$ | Loss tolerance that indicates the convergence |
| `check_waic` | 0.005 | Loss tolerance when we check WAIC |
| . . . and GPLink parameters listed in Table 1. | | |

### A.3    BAYESIAN MODEL SELECTION

In model selection, we are aiming to construct a model with the lowest possible WAIC. Since our copula probability densities are continuous, their values can exceed 1 and the resulting WAIC is typically negative. Zero WAIC corresponds to the Independence model ($\text{pdf} = 1$ on the whole unit square). We also set up a tolerance ($\text{WAIC}_{tol} = 0.005$), and models with $\text{WAIC} \in [-\text{WAIC}_{tol}, \text{WAIC}_{tol}]$ are considered indistinguishable from the independence model.

Since the total number of combinations of 10 copula elements (Fig.1) is large, exhaustive search for the optimal model is not feasible. In our framework, we propose two model algorithms for constructing close-to-optimal copula mixtures: *greedy* and *heuristic*.

### A.4    MODEL SELECTION ALGORITHMS

The greedy algorithm (Algorithm 1) starts by comparing WAIC of all possible single-copula models (from Table 1, in all rotations) and selecting the model with the lowest WAIC. After that, we add one more copula (from another family or in another rotation) to the first selected copula, and prepend the element that yields the lowest WAIC of the mixture. We repeat the process until the WAIC stops decreasing. After the best model is selected, we remove the inessential elements using the `reduce(.)` function. This function removes those elements which have an average concentration of $< 10\%$ everywhere on $x \in [0, 1]$. This step is added to improve the interpretability of the models and computation time for entropy estimation (at a small accuracy cost) and can, in principle, be omitted.

The greedy algorithm can be improved by adding model reduction after each attempt to add an element. In this case, the number of elements can increase and decrease multiple times during the model selection process, which also must be terminated if the algorithm returns to the previously observed solution. Even though it

---

**Algorithm 1:** Greedy algorithm for copula mixture selection

---

1 $M, M_{\text{old}} \leftarrow [\,], [\,]$;
2 $S_c \leftarrow [\texttt{Independence}, \texttt{Gauss}, \texttt{Frank}, 4 \times \texttt{Clayton}, 4 \times \texttt{Gumbel}]$;
            `// 4× includes all rotations`
  `// while every update of the model yields a new best`
3 **while** $\text{WAIC}(M) \leq \text{WAIC}(M_{old})$ **and** $\texttt{size}(S_c) > 0$ **do**
4    $M_{\text{old}} \leftarrow M$ ;
5    select $c$ from $S_c$ such that $\text{WAIC}(\texttt{prepend}(c, M))$ is minimal;
6    $M \leftarrow \texttt{prepend}(c, M)$ ;
7    remove $c$ from $S_c$;
8 **end**
9 $M_{\text{best}} \leftarrow \texttt{reduce}(M_{\text{old}})$;
10 **return** $M_{\text{best}}$;

---

**Algorithm 2:** Heuristic algorithm for copula mixture selection

---

1 $G \leftarrow [\texttt{Gauss}]$;
2 **if** $\text{WAIC}(G) > -\texttt{waic\_tol}$ **then**
3    **return** $[\texttt{Independence}]$;
4 **end**
5 $M_{Cl} \leftarrow [\texttt{Independence}, \texttt{Gauss}, 4 \times \texttt{Clayton}]$;
6 $M_{Gu} \leftarrow [\texttt{Independence}, \texttt{Gauss}, 4 \times \texttt{Gumbel}]$;
7 $M_{\text{best}}, M_{\text{worst}} \leftarrow (M_{Cl}, M_{Gu})$ sorted by WAIC;
8 **if** $\text{WAIC}(G) < \text{WAIC}(M_{best})$ **then**
9    **return** $G$;
10 **end**
11 **for** $i \leftarrow 3 \ldots \texttt{size}(M_{best})$ **do**
12    $M \leftarrow M_{\text{best}}$ with $i$-th element replaced by $M_{\text{worst}}[i]$;
13    $M_{\text{best}} \leftarrow M$ **if** $\text{WAIC}(M) < \text{WAIC}(M_{\text{best}})$;
14 **end**
15 $M_{\text{best}} \leftarrow \texttt{reduce}(M_{\text{best}})$;
16 **if** $\texttt{Gauss} \in M_{best}$ **then**
17    $M \leftarrow M_{\text{best}}$ with $\texttt{Gauss}$ replaced by $\texttt{Frank}$;
18    $M_{\text{best}} \leftarrow M$ **if** $\text{WAIC}(M) < \text{WAIC}(M_{\text{best}})$;
19 **end**
  `// Gauss often gets confused with pairs of e.g.`
     `Clayton 0° + Gumbel 0°`
20 **if** $\texttt{size}(M_{best}) > 1$ **then**
21    **for** $i \leftarrow 1 \ldots (\texttt{size}(M_{best}) - 1)$ **do**
22       **for** $j \leftarrow (i+1) \ldots \texttt{size}(M_{best})$ **do**
23          $M \leftarrow M_{\text{best}}$ with $i$-th and $j$-th elements removed;
24          $M \leftarrow \texttt{prepend}(\texttt{Gauss}, M)$;
25          **if** $\text{WAIC}(M) < \text{WAIC}(M_{best})$ **then**
26             $M_{\text{best}} \leftarrow M$ ;
27             break;
28          **end**
29       **end**
30    **end**
31 **end**
32 $M_{\text{best}} \leftarrow \texttt{reduce}(M_{\text{best}})$;
33 **return** $M_{\text{best}}$;

---

complicates the algorithm, it reduces the maximal execution time (observed on the real neuronal data) from $\sim$90 minutes down to $\sim$40 minutes.

The heuristic algorithm focuses on the tail dependencies (Algorithm 2). First, we try a single Gaussian copula. If variables are not independent, we next compare 2 combinations of 6 elements, which are organized as follows: an Independence copula together with a Gaussian copula and either 4 Clayton or 4 Gumbel copulas in all 4 rotations ($0°$, $90°$, $180°$, $270°$). We select the combination with the lowest WAIC. After that, we take the remaining Clayton/Gumbel copulas one by one and attempt to switch the copula type (Clayton to Gumbel or vise versa). If this switching decreases the WAIC, we keep a better copula type for that rotation and proceed to the next element.

Here we make the assumption, that because Clayton and Gumbel copulas have most of the probability density concentrated in one corner of the unit square (the heavy tail), we can choose the best model for each of the 4 corners independently. When the best combination of Clayton/Gumbel copulas is selected, we can (optionally) reduce the model.

We have not yet used a Frank copula in a heuristic algorithm. We attempt to substitute the Gaussian copula with a Frank copula (if it is still a part of the reduced mixture, see lines 16-19 in Alg. 2). Sometimes, a Gaussian copula can be mistakenly modeled as a Clayton & Gumbel or two Gumbel copulas. So, as a final step (lines 20-31, Alg. 2), we select all pairwise combinations of the remaining elements, and attempt to substitute each of the pairs with a Gaussian copula, selecting the model with the lowest WAIC. Despite a large number of steps in this algorithm, the selection process takes only up to 25 minutes (in case all elements in all rotations are required).

The procedure was designed after observing the model selection process on a variety of synthetic and real neuronal datasets.

## A.5 VINE COPULAS

Vine models provide a way to factorize the high-dimensional copula probability density into a hierarchical set of bivariate copulas (Aas et al., 2009). There are many possible decompositions based on different assumptions about conditional independence of specific elements in a model, which can be classified using graphical models called *regular vines* (Bedford and Cooke, 2001; 2002). A regular vine can be represented using a hierarchical set of trees, where each node corresponds to a conditional distribution function (e.g. $F(u_2|u_1)$) and each edge corresponds to a bivariate copula (e.g. $c(u_2, u_3|u_1)$). The copula models from the lower trees are used to obtain new conditional distributions (new nodes) with additional conditional dependencies for the higher trees, e.g. a ccdf of a copula $c(u_2, u_3|u_1)$ and a marginal conditional distribution $F(u_2|u_1)$ from the 1st tree provide a new conditional distribution $F(u_3|u_1, u_2)$ for a 2nd tree. Therefore, bivariate copula parameters are estimated sequentially, starting from the lowest tree and moving up the hierarchy. The total number of edges in all trees (= the number of bivariate copula models) for an $m$-dimensional regular vine equals $m(m-1)/2$.

The regular vines often assume that the conditional copulas $c(u_i, u_j|\{u_k\})$ themselves are independent of their conditioning variables $\{u_k\}$, but depend on the them indirectly through the conditional distribution functions (nodes) (Acar et al., 2012). This is known as the *simplifying assumption* for vine copulas (Haff et al., 2010), which, if applicable, allows to escape the curse of dimensionality in high-dimensional copula construction.

In this study, we focus on the *canonical vine* or *C-vine*, which has a unique node in each tree, connected to all of the edges in that tree. For illustration, see, for example, Figure 2 in Aas et al. (2009). The C-vine was shown to be a good choice for neuronal datasets (Onken & Panzeri, 2016), as they often include some proxy of neuronal population activity as an outstanding variable, strongly correlated with the rest. This variable provides a natural choice for the first conditioning variable in the lowest tree. In the neuronal datasets from Henschke et al. (2020), this outstanding variable is the global fluorescence signal in the imaged field of view (global neuropil).

To construct a C-vine for describing the neuronal and behavioral data from Henschke et al. (2020), we used a heuristic element ordering based on the sum of absolute values of Kendall's $\tau$ of a given element with all of the other elements. It was shown by Czado et al. (2012) that this ordering facilitates C-vine modeling. For all of the animals and most of the recordings (14 out of 16), including the one used in Figure 3, the first variable after such ordering was the global neuropil activity. This again confirms, that a C-vine with the global neuropil activity as a first variable is an appropriate model for the dependencies in neuronal datasets.

## A.6 ALGORITHMIC COMPLEXITY

In this section, we discuss the algorithmic complexity of the parameter inference for a C-vine copula model.

The parameter inference for each of the bivariate Copula-GP models scales as $\mathcal{O}(n)$, where $n$ is the number of samples, since we use a scalable kernel interpolation KISS-GP (Wilson and Nickisch, 2015). As we mentioned in Sec. A.5, a full $m$-dimensional C-vine model requires $m(m-1)/2$ bivariate copulas, trained sequentially.

As a result, the $\mathcal{O}(n)$ GP parameter inference has to be repeated $m(m-1)/2$ times, which yields $\mathcal{O}(n \cdot m^2)$ complexity.

In practice, the computational cost (in terms of time) of the parameter inference for each bivariate model varies from tens of seconds to tens of minutes. The heuristic model selection is designed in such a way, that it discards independent variables in just around 20 seconds (line 3 in Alg. 2). As a result, most of the models are quickly skipped and further considered as Independence models, and their contribution to the total computational cost can be neglected. When the model is evaluated, the Independence components are also efficiently 'skipped' during sampling, as `ppcf` function is not called for them. The Independence models also add zero to C-vine log probability, so they are also 'skipped' during log probability calculation. They also reduce the total memory storage, as no GP parameters, which predominate the memory requirements, are stored for these models.

In a conditional C-vine trained on a real neuronal dataset with 109 variables, 5253 out of 5886 (89%) bivariate models were Independence, which leaves only 633 non-Independence models.

In practice, this means that the algorithmic complexity of the model is much better than the naïve theoretical prediction $\mathcal{O}(n \cdot m^2)$, based on the structure of the graphical model. Suppose that the actual number of the non-Independence models $N_{nI}$ in a vine model is much smaller than $m(m-1)/2$ and can be characterized by an effective number of dimensions $m_{eff} \sim \sqrt{N_{nI}}$. In this case, instead of the $\mathcal{O}(m^2)$ scaling with the number of variables, the complexity highly depends on the sparsity of the dependencies in the graphical model and scales with as $\mathcal{O}(n \cdot N_{nI}) \sim \mathcal{O}(n \cdot m_{eff}^2)$.

Therefore, the our method is especially efficient on the datasets with a low effective dimensionality $m_{eff}$, such as the neuronal data. The number of variables $m$ itself has little effect on the computational cost and memory storage.

## B    MORE VALIDATION ON SYNTHETIC DATA

**Computing infrastructure**    We developed our framework and ran the majority of our experiments (described both in the paper and Supplemental Material) on an Ubuntu 18.04 LTS machine with 2 x Intel(R) Xeon(R) Gold 6142 CPU @ 2.60GHz and 1x GeForce RTX 2080 + 1 x GeForce RTX 2080 Ti GPUs. For training C-vine models, we used another Scientific Linux 7.6 machine with 1 x Intel(R) Xeon(R) Silver 4114 CPU @ 2.20GHz and 8 x GeForce RTX 2080 Ti GPUs.

**Code availability**    Code will be made available on GitHub upon paper acceptance.

### B.1    MODEL SELECTION FOR BIVARIATE COPULAS

**Synthetic data**    We generate artificial data by sampling from a copula mixture, parametrized in two different ways:

1. mixing concentrations of all copulas were constant and equal to $1/N$ ($N$ = number of copulas), but copula parameters $\theta$ were parametrized by the phase-shifted sinus functions:

$$\theta_i = A_i \sin\left(\pi m \frac{i}{N} + 2\pi x\right) + B_i, \qquad x \in [0,1] \tag{A4}$$

   where $i$ is the index of the copula in a mixture, $m = 1$. For Clayton and Gumbel copulas, the absolute value of the sinus was used. The amplitudes $A_i$ were chosen to cover most of the range of parameters, except for extremely low or high $\theta$s for which all copula families become indistinguishable (from independence or deterministic dependence, respectively).

2. copula parameters $\theta$ were constant, but mixing concentrations $\phi$ were parametrized by the phase-shifted sinus functions (same as Eq. A4, with $A_i = B_i = 1/N$ and $m = 2$). Such parametrization ensures that the sum of all mixing concentrations remains equal to one ($\sum_{i=1}^{N} \phi = 1$). Yet, each $\phi$ turns to zero somewhere along this trajectory, allowing us to discriminate the models and infer the correct mixture.

**Identifiability tests**    We tested the ability of the model selection algorithms to select the correct mixture of copula models, the same as the one from which the data was generated. We generated 5000 samples with equally spaced unique inputs on [0,1].

Both model selection algorithms were able to correctly select all of the 1-component and most of the 2-component models on simulated data. For simulated data with larger numbers of components (or 2 very similar components), the WAIC of the selected model was either lower (which is possible given a limited number of samples) or close to the WAIC of the correct parametric model. In other words, the difference between the

WAIC of the correct model and of the best selected model never exceeded the $\text{WAIC}_{test\_tol} = 0.05$, which we set up as a criteria for passing the test: $\Delta\text{WAIC} < \text{WAIC}_{test\_tol}$. Since all the tests were passed successfully, we conclude that both algorithms are capable of finding optimal or close-to-optimal solutions for copula mixtures.

**A more detailed report on the model identifiability tests** Tables A3-A7 below illustrate the search for the best model. The copula model names in these tables are shortened to the first two letters, e.g. Gumbel becomes 'Gu', Frank becomes 'Fr'. The information in these Tables provides some intuition on the model selection process and the range of WAICs for the correct or incorrect models. The final selected models are shown in bold.

Table A3 demonstrates that both greedy and heuristic algorithms can identify the correct single copula model. Some key intermediate models ($M$ in Alg. 1-2) with their WAICs are listed in the table, along with the total duration of simulations (T, in minutes) on RTX 2080Ti for both algorithms.

Table A4 shows the identification of the mixtures with 2 components, where the copula parameters $\theta$ were constant (independent of $x$) and mixing concentrations $\phi$ were parameterized by the phase-shifted sinus functions (Eq. A4). All of these models were correctly identified with both algorithms. The mixtures with 2 components, where the copula parameters $\theta$ varied harmonically (as in Eq. A4) but the mixing concentrations $\phi$ were constant, were harder to identify. Table A5 shows that a few times, each of the algorithms selected a model that was better than the true model ($\text{WAIC}_{best} - \text{WAIC}_{true} < 0$). The greedy algorithm made one mistake, yet the model it selected was very close to optimal. Such misidentification happens due to the limited number of samples in a given synthetic dataset.

Tables A6-A7 show the model selection for 3 component models. Again, as in Tables A4-A5, either $\theta$ or $\phi$ was constant. Here, the model selection algorithms could rarely identify the correct model (due to overcompleteness of the mixture models), but always selected the one that was very close to optimal: $\text{WAIC}_{best} - \text{WAIC}_{true} \ll \text{WAIC}_{test\_tol}$.

Note, that $\text{WAIC}_{test\_tol}$ is different from `waic_tol`. We have set `waic_tol` for comparison against Independent model to such a small value (10x smaller than $\text{WAIC}_{test\_tol}$) because we want to avoid making false assumptions about conditional independences in the model. Also note, that the WAIC of the true model depends on the particular synthetic dataset generated in each test. Therefore, the final WAIC in the left and in the right columns of Tables A3-A7 can be slightly different (yet, right within $\text{WAIC}_{test\_tol}$).

## B.2 ACCURACY OF ENTROPY ESTIMATION

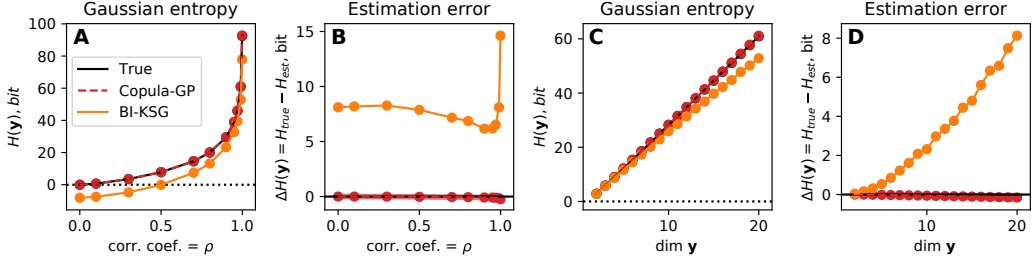

Figure A4: Accuracy of the entropy estimation for multivariate Gaussian distributions. **A** Entropy of the 20-dimensional multivariate Gaussian distribution for different correlation coefficients $\rho$. **B** Estimation error for the entropy shown in A. **C** Entropy of the multivariate Gaussian distributions with $\rho = 0.99$ and varying dimensionality. **D** Estimation error for the entropy shown in C.

In this section, we consider a fixed copula mixture model with known parameters $\theta$ and test the reliability of the entropy estimation with Monte Carlo (MC) integration. We test the accuracy of the entropy estimation on a multivariate Gaussian distribution, with $\text{cov}(y_i, y_j) = \rho + (1 - \rho)\delta_{ij}$, where $\delta_{ij}$ is Kronecker's delta and $\rho \in [0, 0.999]$. Given a known Gaussian copula, we estimate the entropy with MC integration and compare it to the analytically calculated true value. We set up a tolerance to $\Delta H = 0.01(\dim \mathbf{y})$. As a result, for every correlation $\rho$ (Fig. A4A-B) and every number of dimensions $\dim \mathbf{y}$ (Fig. A4C-D), the Copula-GP provides an accurate result, within the error margin. In Figure A4, BI-KSG estimates (Gao et al., 2016) obtained on the dataset with 10k samples are shown for comparison. This experiment 1) validates the MC integration; 2) validates the numerical stability of the probability density function of the Gaussian copula up to a specified maximal $\rho = 0.999$ (for $\rho > 0.999$ the model is indistinguishable from the deterministic dependence $u_1 = u_2$).

### B.3 VALIDATION ON A NON-GAUSSIAN COPULA

Copula-GP is guaranteed to produce unbiased entropy estimates when the true dependency matches the parametric model (i.e. is Gaussian, Clayton, Frank or Gumbel, or any linear mixture of those), assuming that the implementation of log-likelihood and sampling is correct, which is ensured by the tests and correctly estimated parameters.

Figure 2A tested the entropy and mutual information estimates on the Gaussian copula. Here, we follow the same procedure and generate samples from a C-Vine, in which each bivariate element is a Clayton copula. We varied the parameter $\theta$ of the Clayton copula linearly from 0 to 2.

The observed result is qualitatively identical to the validation on a Gaussian copula model (see Fig. 2A): Copula-GP measures both entropy and mutual information exactly (within integration tolerance, see Fig. A5), while the performance of the non-parametric methods on this dataset is lower.

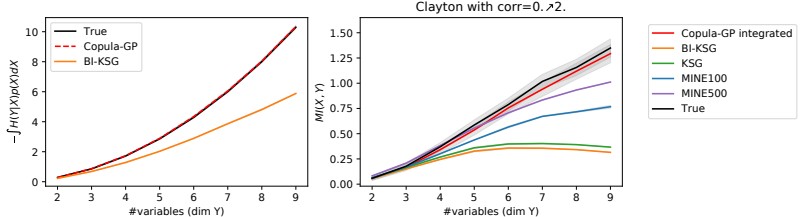

Figure A5: Conditional entropy $H(\mathbf{y}|x)$ (left panel) and mutual information $I(x, \mathbf{y})$ (right panel) measured by different methods on synthetic data, generated from Clayton copula. The performance of various estimators is qualitatively similar to Fig. 2A with the Gaussian copula.

### B.4 PERFORMANCE ON A UCI BENCHMARK DATASET

Here we compare our method with the similar method from Lopez-Paz et al. (2013) and Hernández-Lobato et al. (2013). Apart from differences in implementation (SVI vs. EP, (un)/conditional marginals), our Copula-GP method has two conceptual improvements 1) we use non-parametric **conditional** marginals, instead of unconditional; 2) we use linear mixtures of copula elements with qualitatively different tail dependencies. Using the UCI *shuttle* dataset (Frank et al., 2011), we compare the performance of the Lopez-Paz model vs. Copula-GP model which was either allowed to use only one copula element, or a mixture of any number of elements (selected with the model selection method, see Appx. A.4). We used vine copulas truncated at 3 or 5 trees, ran them on the training set and evaluated the log-likelihood on the test set (higher is better).

As a result, Copula-GP restricted to the use of only one copula element produced similar results to Lopez-Paz et al. (2013), while our mixture model considerably outperformed both of these single element models.

Table A2: Average test log-likelihood for Copula-GP vs. Lopez-Paz et al. (2013)

| Trees | Lopez-Paz et al. | Single copula | Mixture |
|---|---|---|---|
| 3 | 3.645±0.427 | 3.759±0.307 | **4.326±0.170** |
| 9 | 4.755±0.389 | 4.470±0.215 | **5.115±0.150** |

## C INTERPRETATION OF COPULA-GP MODELS

### C.1 CONDITIONAL VS. UNCONDITIONAL MARGINALS

In this section, we explain the difference between copula models with conditional vs. unconditional marginals.

Consider a pair of neurons, which both fire strongly in a certain context (e.g. in the reward zone of the VR environment, see Fig. A6A). We simulate the activity of these neurons using a Poisson model (Fig. A6B) and then convolve the simulated spike count data with the exponential kernel, in order to generate continuous data similar to the recordings shown in Fig. 3. The resulting generated activity is shown in Fig. A6C.

Let us then use unconditional marginals to project simulated neuronal recordings onto a unit cube (Fig. A6D, left). The color of the data point shows the position $X$, and one can see that the orange and red points ($X > 0.5$) are concentrated in the upper right corner of the plot. If we plot a histogram for one of the neurons and only

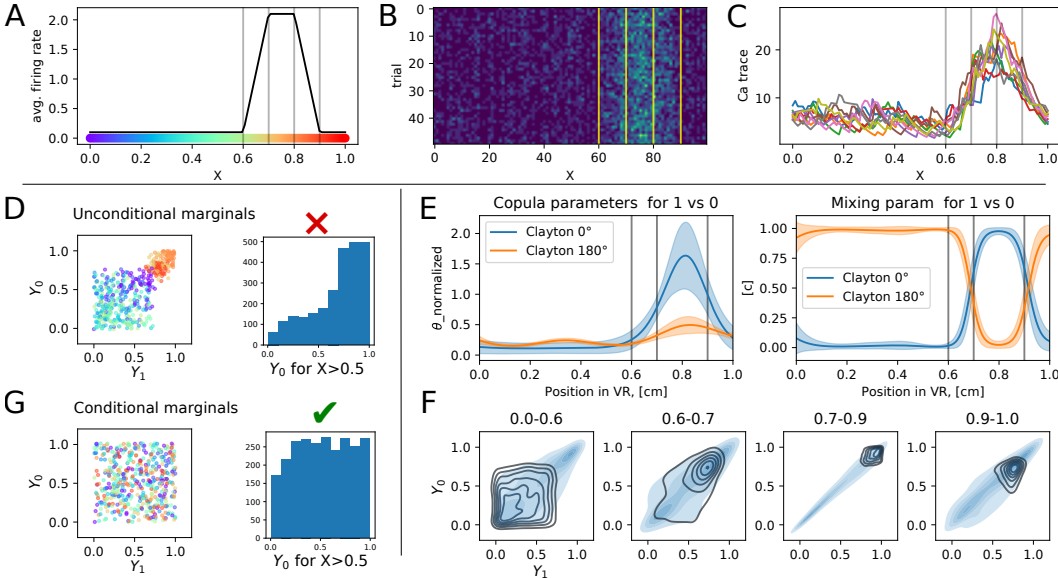

Figure A6: Toy model of two independent neurons with the same conditional firing rate: **A** average firing rate depending on $X$; **B** raster plot with spike counts for one of the neurons; **C** simulated calcium traces from 10 trials for one of the neurons; **D** data-points projected onto a unit cube by the probability integral transform based on unconditional marginals; colored points show joint neural data-point $(Y_0, Y_1)$ recorded at $X$, where $X$ is color-coded (as in A); the histogram on the right shows conditional marginal distribution $p(Y_0 | X > 0.5)$; **E** mixture model fit for the data in D (with unconditional marginals); **F** density plots comparing predictions of the mixture model (shades) with the actual data (black outlines); the fit is poor due to non-uniformity of empirical marginal distributions; **G** same as D, but based on conditional marginals, estimated using fastKDE.

orange-red points, which corresponds to $p(Y_i | X > 0.5)$, we see that it is far from being uniform (Fig. A6D, right).

In previous works on conditional copulas (Hernández-Lobato et al., 2013), the marginals were assumed to be unconditional. Such statistical copula-based models would ignore the non-uniformity of the empirical copula marginals, and still attempt to describe the data with some heavy-tailed copula (Fig. A6E). However, uniformity of copula marginals is one of the main model assumptions, and it does not hold in this case. As a result, the model fit is poor (Fig. A6F).

Modelling with conditional marginals, as in our Copula-GP model, is radically different. The non-parametric conditional marginal model (estimated with fastKDE) reflects the fact that both neurons are typically firing strongly around $X \approx 0.7$, and would transform the marginals accordingly (Fig. A6G, left). As a result, all the model assumptions, including uniformity of copula marginals, will strictly hold (Fig. A6G, right). Additionally, our copula model would focus solely on the joint trial-to-trial variability (i.e. on noise correlations) of these neurons, regardless of the absolute values of their firing rates. In this example, the model finds no noise correlations (best fit for Fig. A6G is Independence), which is the correct result for the simulation of two independent neurons.

Therefore, conditional marginals make our model more flexible and capable of holding the modelling assumptions even on complex neuronal data with highly variable marginal statistics. Also, for the neuronal data in particular, the model becomes more interpretable, as the conditional copula $c(\mathbf{u}^{\mathbf{x}} | x)$ focuses only on the joint trial-to-trial variability of neural activity. Therefore, the statistics corresponding to the network becomes entirely separated from the single unit statistics. This makes our Copula-GP model particularly well suited for neuroscience applications.

## C.2 Model parameters for the bivariate neuronal and behavioral examples

In this section, we provide visualisations for the parameters of the bivariate copula models from Figure 3C-F and discuss the interpretability of these models.

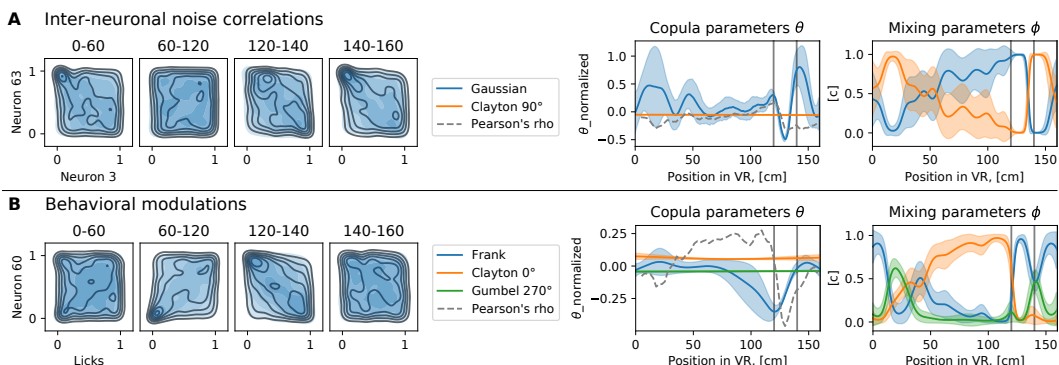

Figure A7: Parameters of the copula mixture models. From left to right: copula probability densities (same as Fig.3C-D); a list of selected copula elements; copula parameters $\theta$; mixing concentrations $\phi$. These plots are provided for: **A** the noise correlation example; **B** the behavioral modulation example.

Figure A7 shows the probability density of the joint distribution of two variables and the parameters of a corresponding Copula-GP mixture model. The plots on the left repeat Fig.3C-D and represent the true density (outlines) and the copula model density (blue shades) for each part of the task.

In the noise correlation example (Fig. A7A), we observe the *tail dependencies* between the variables (i.e. concentration of the probability density in a corner of the unit square) around [0-60] cm and [140-160] cm of the virtual corridor. There is only one element with a tail dependency in this mixture: Clayton $90°$ copula. On the right-most plot in Fig. A7A, we see the mixing concentration for the elements of the mixture model. The concentration of Clayton $90°$ copula (orange line) is close to 100% around 20 cm and 150 cm, which agrees with our observations from the density plots.

The confidence intervals ($\pm 2\sigma$) for the parameters approximated with Gaussian processes are shown with shaded areas in parameter plots. These intervals provide a measure of uncertainty in model parameters. For instance, when the concentration of the Gaussian copula in the mixture is close to 0% ($x$ around 20 cm and 150 cm), the confidence intervals for the Gaussian copula parameter ($\theta$, blue shade) in Fig. A7A become very wide (from almost 0 to 1). Since this copula element is not affecting the mixture for those values of $x$, its $\theta$ parameter has no effect on the mixture model log probability. Therefore, this parameter is not constrained to any certain value. In a similar manner, we see that the variables are almost independent between 60 and 120 cm (see density plots on the left in Fig. A7). Both copula elements can describe this independence. As a result, the mixing concentrations for both elements have high uncertainty in that interval of $x$. Yet, Gaussian copula with a slightly positive correlation is still a bit more likely to describe the data in that interval.

The copula parameter plot in Fig. A7A also shows Pearson's $\rho$, which does not change much in this example and remains close to zero. This illustrates, that the traditional linear noise correlation analysis would ignore (or downplay) this pair of neurons as the ones with no dependence. This happens because the Pearson's $\rho$ only captures the linear correlation and ignores the tail dependencies, whereas our model provides a more detailed description of the joint bivariate distribution.

In the behavioral modulation example (Fig. A7B), we observe more complicated tail dependencies in the density plots. The best selected model supports this observation and provides a mixture model with 3 components, 2 of which have various tail dependencies. The Clayton $0°$ copula (orange) describes the lower tail dependence observed in the second part of the virtual corridor with gratings (around [60-120] cm, see Fig. 3A for task structure). This dependence can be verbally interpreted as follows: *when there is **no** licking, the Neuron 60 is certainly silent; but when the animal **is** licking, the activity of Neuron 60 is slightly positively correlated with the licking rate*.

These examples illustrate, that by analysing the copula parameters and the mixing concentrations of the Copula-GP mixture model, one can interpret the changes in the bivariate dependence structure. Just like traditional *tuning curves* characterize the response of a single neuron, our mixture model characterizes the 'tuning' of the dependence structure between pairs of variables to a given stimulus or context. Knowing the qualitative properties of the copula elements that constitute a copula mixture, one can focus on the dominant element of the copula mixture for every given conditioning variable $x$ and describe the shape of the dependence.

### C.3 ABLATION STUDY

After the model is selected, one can ablate the elements one by one and check the WAIC of these models. Following this procedure, one can test whether all of the elements are important in the mixture.

We performed the ablation on the models from Fig. A7. All models after ablation had higher WAIC than the originally selected model. Ablation of the Clayton copula in Fig. A7B increased WAIC the most (from -0.043 to -0.033), while both elements in Fig. A7A were equally important (WAIC after ablation was the same (within tolerance)).

### C.4 DETECTION OF HEAVY TAILS

Since the Copula-GP framework is based on a linear mixture model, it is possible to calculate the probability that a certain data-point was generated from a heavy-tailed component of the distribution (e.g. Clayton). Given a mixture model with estimated parameters, described by:

$$c\left(\mathbf{u}|x\right) = \sum_{j=1}^{K} \phi_j(x)c_j(\mathbf{u};\theta_j(x)),$$

where $u_i = \text{CDF}_i(y_i|x)$, $x$ is a conditioning variable and $\mathbf{y}$ is a vector of neuronal and/or behavioral recordings. Then, for a data-point $(x, \mathbf{y})$, this probability is equal to:

$$p(\text{clayton}|x, \mathbf{y}) = \frac{p(\mathbf{y}|\text{clayton}, x) \cdot p(\text{clayton}|x)}{p(\mathbf{y}|x)} = \frac{\sum_{j\in\text{Clayton}} \phi_j(x)c_j(\mathbf{u};\theta_j(x))}{\sum_{j\in\text{All}} \phi_j(x)c_j(\mathbf{u};\theta_j(x))}.$$

By thresholding $p(\text{clayton}|x, \mathbf{y})$ and $p(\mathbf{y}|x)$, one can select those data-points that constitute the heavy tail of the distribution, which is described by Clayton copulas in our copula mixture model.

## APPENDIX REFERENCES

James Hensman, Alexander Matthews, and Zoubin Ghahramani. Scalable variational gaussian process classification. 2015.

Andrew Wilson and Hannes Nickisch. Kernel interpolation for scalable structured Gaussian processes (KISS-GP). In *International Conference on Machine Learning*, pages 1775–1784, 2015.

Christopher KI Williams and Carl Edward Rasmussen. *Gaussian processes for machine learning*, volume 2. MIT press Cambridge, MA, 2006.

Tim Bedford and Roger M Cooke. Probability density decomposition for conditionally dependent random variables modeled by vines. *Annals of Mathematics and Artificial intelligence*, 32(1-4):245–268, 2001.

Tim Bedford and Roger M Cooke. Vines: A new graphical model for dependent random variables. *Annals of Statistics*, pages 1031–1068, 2002.

Elif F Acar, Christian Genest, and Johanna NešLehová. Beyond simplified pair-copula constructions. *Journal of Multivariate Analysis*, 110:74–90, 2012.

Ingrid Hobæk Haff, Kjersti Aas, and Arnoldo Frigessi. On the simplified pair-copula construction—simply useful or too simplistic? *Journal of Multivariate Analysis*, 101(5):1296–1310, 2010.

David Lopez-Paz, Jose Miguel Hernández-Lobato, and Ghahramani Zoubin. Gaussian process vine copulas for multivariate dependence. In *International Conference on Machine Learning*, pages 10–18, 2013.

José Miguel Hernández-Lobato, James R Lloyd, and Daniel Hernández-Lobato. Gaussian process conditional copulas with applications to financial time series. In C. J. C. Burges, L. Bottou, M. Welling, Z. Ghahramani, and K. Q. Weinberger, editors, *Advances in Neural Information Processing Systems 26*, pages 1736–1744. Curran Associates, Inc., 2013. URL http://papers.nips.cc/paper/5084-gaussian-process-conditional-copulas-with-applications-to-financial-time-series.pdf.

Andrew Frank, Arthur Asuncion, et al. UCI machine learning repository, 2010. *URL http://archive. ics. uci. edu/ml*, 15:22, 2011.

Table A3: The model selection histories for 1-element mixtures

| True Model | Greedy | | | Heuristic | | |
|---|---|---|---|---|---|---|
| | Search attempts | WAIC | T | Search attempts | WAIC | T |
| $Ga$ | $Ga$ | -0.1619 | 25 m | $Ga$ | -0.1513 | 3 m |
| | $GaFr$ | -0.1610 | | $InGaGu^{180}Gu^{270}Gu^0Gu^{90}$ | -0.1499 | |
| | **$Ga$** | **-0.1619** | | $InGaCl^0Cl^{90}Cl^{180}Cl^{270}$ | -0.1498 | |
| | | | | **$Ga$** | **-0.1513** | |
| $Fr$ | $Fr$ | -0.1389 | 57 m | $Ga$ | -0.1400 | 3 m |
| | $FrCl^{90}$ | -0.1395 | | $InGaGu^{180}Gu^{270}Gu^0Gu^{90}$ | -0.1391 | |
| | $FrCl^{90}Gu^{270}$ | -0.1396 | | $InGaCl^0Cl^{90}Cl^{180}Cl^{270}$ | -0.1391 | |
| | $FrCl^{90}Gu^{270}Gu^{90}$ | -0.1396 | | **$Fr$** | **-0.1509** | |
| | **$Fr$** | **-0.1389** | | | | |
| $Cl^0$ | $Cl^0$ | -0.5225 | 37 m | $Ga$ | -0.3825 | 5 m |
| | $Cl^0Gu^0$ | -0.5226 | | $InGaGu^{180}Gu^{270}Gu^0Gu^{90}$ | -0.4943 | |
| | $Cl^0Gu^0Cl^{180}$ | -0.5225 | | $InGaCl^0Cl^{90}Cl^{180}Cl^{270}$ | -0.5303 | |
| | **$Cl^0$** | **-0.5224** | | **$Cl^0$** | **-0.5311** | |
| $Gu^0$ | $Gu^0$ | -0.6267 | 43 m | $Ga$ | -0.5555 | 7 m |
| | $Gu^0Cl^{180}$ | -0.6268 | | $InGaGu^{180}Gu^{270}Gu^0Gu^{90}$ | -0.5988 | |
| | $Gu^0Cl^{180}Gu^{180}$ | -0.6267 | | $InGaCl^0Cl^{90}Cl^{180}Cl^{270}$ | -0.5946 | |
| | **$Gu^0$** | **-0.6230** | | $GaGu^0$ | -0.6040 | |
| | | | | **$Gu^0$** | **-0.6050** | |
| $Cl^{90}$ | $Cl^{90}$ | -0.5389 | 22 m | $Ga$ | -0.3922 | 5 m |
| | $Cl^{90}Cl^{270}$ | -0.5389 | | $InGaGu^{180}Gu^{270}Gu^0Gu^{90}$ | -0.5047 | |
| | **$Cl^{90}$** | **-0.5389** | | $InGaCl^0Cl^{90}Cl^{180}Cl^{270}$ | -0.5409 | |
| | | | | **$Cl^{90}$** | **-0.5410** | |
| $Gu^{90}$ | $Gu^{90}$ | -0.6137 | 55 m | $Ga$ | -0.5501 | 7 m |
| | $Gu^{90}Gu^{270}$ | -0.6144 | | $InGaGu^{180}Gu^{270}Gu^0Gu^{90}$ | -0.5893 | |
| | $Gu^{90}Gu^{270}Cl^{270}$ | -0.6145 | | $InGaCl^0Cl^{90}Cl^{180}Cl^{270}$ | -0.5831 | |
| | $Gu^{90}Gu^{270}Cl^{270}Cl^{90}$ | -0.6144 | | $GaGu^{90}$ | -0.5887 | |
| | **$Gu^{90}$** | **-0.6137** | | **$Gu^{90}$** | **-0.5950** | |
| $Cl^{180}$ | $Cl^{180}$ | -0.5566 | 36 m | $Ga$ | -0.3932 | 7 m |
| | $Cl^{180}Cl^0$ | -0.5582 | | $InGaGu^{180}Gu^{270}Gu^0Gu^{90}$ | -0.4956 | |
| | $Cl^{180}Cl^0In$ | -0.5582 | | $InGaCl^0Cl^{90}Cl^{180}Cl^{270}$ | -0.5493 | |
| | **$Cl^{180}$** | **-0.5565** | | **$Cl^{180}$** | **-0.5489** | |
| $Gu^{180}$ | $Gu^{180}$ | -0.6131 | 43 m | $Ga$ | -0.5553 | 6 m |
| | $Gu^{180}Cl^0$ | -0.6164 | | $InGaGu^{180}Gu^{270}Gu^0Gu^{90}$ | -0.6091 | |
| | $Gu^{180}Cl^0Fr$ | -0.6163 | | $InGaCl^0Cl^{90}Cl^{180}Cl^{270}$ | -0.6045 | |
| | **$Gu^{180}$** | **-0.6131** | | **$Gu^{180}$** | **-0.6154** | |
| $Cl^{270}$ | $Cl^{270}$ | -0.5434 | 23 m | $Ga$ | -0.3909 | 5 m |
| | $Cl^{270}Gu^{270}$ | -0.5433 | | $InGaGu^{180}Gu^{270}Gu^0Gu^{90}$ | -0.5094 | |
| | **$Cl^{270}$** | **-0.5434** | | $InGaCl^0Cl^{90}Cl^{180}Cl^{270}$ | -0.5535 | |
| | | | | **$Cl^{270}$** | **-0.5548** | |
| $Gu^{270}$ | $Gu^{270}$ | -0.5928 | 51 m | $Ga$ | -0.5763 | 6 m |
| | $Gu^{270}Cl^{90}$ | -0.5934 | | $InGaGu^{180}Gu^{270}Gu^0Gu^{90}$ | -0.6277 | |
| | $Gu^{270}Cl^{90}In$ | -0.5935 | | $InGaCl^0Cl^{90}Cl^{180}Cl^{270}$ | -0.6179 | |
| | $Gu^{270}Cl^{90}InCl^{180}$ | -0.5931 | | **$Gu^{270}$** | **-0.6300** | |
| | **$Gu^{270}$** | **-0.5928** | | | | |

Table A4: The model selection histories for 2-element mixtures with constant $\theta$ and variable $\phi$

| True Model | Greedy | | | Heuristic | | |
|---|---|---|---|---|---|---|
| | Search attempts | WAIC | T | Search attempts | WAIC | T |
| $Gu^{90}$ $Ga$ | Ga | -0.1877 | 101 m | Ga | -0.1922 | 11 m |
| | $GaGu^{90}$ | -0.2855 | | $InGaGu^{180}Gu^{270}Gu^{0}Gu^{90}$ | -0.3070 | |
| | $GaGu^{90}Cl^{270}$ | -0.2855 | | $InGaCl^{0}Cl^{90}Cl^{180}Cl^{270}$ | -0.2996 | |
| | $GaGu^{90}Cl^{270}Fr$ | -0.2856 | | $GaCl^{0}Gu^{0}Gu^{90}$ | -0.3082 | |
| | $GaGu^{90}Cl^{270}FrGu^{270}$ | -0.2856 | | $GaCl^{0}Cl^{180}Gu^{90}$ | -0.3076 | |
| | $GaGu^{90}Cl^{270}FrGu^{270}$-$Cl^{90}$ | -0.2856 | | **$GaGu^{90}$** | **-0.3091** | |
| | **$Gu^{90}Ga$** | **-0.2854** | | | | |
| $Ga$ $Cl^{270}$ | Fr | -0.1635 | 87 m | Ga | -0.1600 | 5 m |
| | $FrCl^{270}$ | -0.2707 | | $InGaGu^{180}Gu^{270}Gu^{0}Gu^{90}$ | -0.2687 | |
| | $FrCl^{270}Ga$ | -0.2747 | | $InGaCl^{0}Cl^{90}Cl^{180}Cl^{270}$ | -0.2835 | |
| | $FrCl^{270}GaGu^{180}$ | -0.2782 | | **$GaCl^{270}$** | **-0.2845** | |
| | $FrCl^{270}GaGu^{180}Cl^{90}$ | -0.2781 | | | | |
| | **$GaCl^{270}$** | **-0.2821** | | | | |
| $Gu^{180}$ $Fr$ | $Gu^{180}$ | -0.1681 | 99 m | Ga | -0.1534 | 8 m |
| | $Gu^{180}Fr$ | -0.2099 | | $InGaGu^{180}Gu^{270}Gu^{0}Gu^{90}$ | -0.1993 | |
| | $Gu^{180}FrCl^{180}$ | -0.2101 | | $InGaCl^{0}Cl^{90}Cl^{180}Cl^{270}$ | -0.1977 | |
| | $Gu^{180}FrCl^{180}Cl^{90}$ | -0.2105 | | $InGaGu^{180}$ | -0.2074 | |
| | $Gu^{180}FrCl^{180}Cl^{90}In$ | -0.2106 | | **$FrGu^{180}$** | **-0.2104** | |
| | $Gu^{180}FrCl^{180}Cl^{90}In$-$Gu^{270}$ | -0.2099 | | | | |
| | **$FrGu^{180}$** | **-0.2099** | | | | |
| $Cl^{0}$ $Cl^{90}$ | Fr | -0.1587 | 92 m | Ga | -0.1652 | 5 m |
| | $FrCl^{0}$ | -0.2600 | | $InGaGu^{180}Gu^{270}Gu^{0}Gu^{90}$ | -0.3142 | |
| | $FrCl^{0}Cl^{90}$ | -0.3173 | | $InGaCl^{0}Cl^{90}Cl^{180}Cl^{270}$ | -0.3430 | |
| | $FrCl^{0}Cl^{90}Gu^{270}$ | -0.3176 | | **$Cl^{0}Cl^{90}$** | **-0.3448** | |
| | $FrCl^{0}Cl^{90}Gu^{270}In$ | -0.3176 | | | | |
| | $FrCl^{0}Cl^{90}Gu^{270}InCl^{270}$ | -0.3175 | | | | |
| | **$Cl^{90}Cl^{0}$** | **-0.3190** | | | | |
| $Cl^{180}$ $Gu^{270}$ | Fr | -0.2204 | 103 m | Ga | -0.1965 | 7 m |
| | $FrCl^{180}$ | -0.3488 | | $InGaGu^{180}Gu^{270}Gu^{0}Gu^{90}$ | -0.3591 | |
| | $FrCl^{180}Gu^{270}$ | -0.3874 | | $InGaCl^{0}Cl^{90}Cl^{180}Cl^{270}$ | -0.3688 | |
| | $FrCl^{180}Gu^{270}Cl^{90}$ | -0.3877 | | $GaGu^{270}Cl^{180}$ | -0.3771 | |
| | $FrCl^{180}Gu^{270}Cl^{90}Ga$ | -0.3878 | | **$Gu^{270}Cl^{180}$** | **-0.3772** | |
| | $FrCl^{180}Gu^{270}Cl^{90}Ga$-$Gu^{90}$ | -0.3878 | | | | |
| | **$Gu^{270}Cl^{180}$** | **-0.3888** | | | | |

Table A5: The model selection histories for 2-element mixtures with constant $\phi$ and variable $\theta$

| True Model | Greedy | | | Heuristic | | |
|---|---|---|---|---|---|---|
| | Search attempts | WAIC | T | Search attempts | WAIC | T |
| $Gu^{90}$ $Ga$ | $Gu^{90}$ | -0.1419 | 60 m | Ga | -0.1538 | 10 m |
| | $Gu^{90}Fr$ | -0.2022 | | $InGaGu^{180}Gu^{270}Gu^{0}Gu^{90}$ | -0.2320 | |
| | $Gu^{90}FrCl^{270}$ | -0.2024 | | $InGaCl^{0}Cl^{90}Cl^{180}Cl^{270}$ | -0.2218 | |
| | $Gu^{90}FrCl^{270}Ga$ | -0.2024 | | $GaCl^{90}Gu^{0}Gu^{90}$ | -0.2321 | |
| | **$FrGu^{90}$** | **-0.2021** | | **$GaGu^{90}$** | **-0.2326** | |
| | $WAIC_{best} - WAIC_{true}$: | **-0.0013** | | | | |
| $Ga$ $Cl^{270}$ | $Gu^{90}$ | -0.1495 | 56 m | Ga | -0.1062 | 7 m |
| | $Gu^{90}Fr$ | -0.1894 | | $InGaGu^{180}Gu^{270}Gu^{0}Gu^{90}$ | -0.1747 | |
| | $Gu^{90}FrCl^{270}$ | -0.1915 | | $InGaCl^{0}Cl^{90}Cl^{180}Cl^{270}$ | -0.1783 | |
| | $Gu^{90}FrCl^{270}In$ | -0.1902 | | $GaGu^{0}Cl^{270}$ | -0.1812 | |
| | **$Cl^{270}FrGu^{90}$** | **-0.1915** | | **$GaCl^{270}$** | **-0.1801** | |
| | $WAIC_{best} - WAIC_{true}$: | **0.0032** | | | | |

| True Model | Greedy | | | Heuristic | | |
|---|---|---|---|---|---|---|
| | Search attempts | WAIC | T | Search attempts | WAIC | T |
| $Gu^{180}$ Fr | $Gu^{180}$ | -0.1600 | 58 m | Ga | -0.1331 | 8 m |
| | $Gu^{180}Fr$ | -0.2191 | | $InGaGu^{180}Gu^{270}Gu^0Gu^{90}$ | -0.1944 | |
| | $Gu^{180}FrCl^{270}$ | -0.2195 | | $InGaCl^0Cl^{90}Cl^{180}Cl^{270}$ | -0.1936 | |
| | $Gu^{180}FrCl^{270}Cl^0$ | -0.2190 | | $GaGu^{180}Cl^{90}Gu^0Gu^{90}$ | -0.1945 | |
| | **$FrGu^{180}$** | **-0.2190** | | **$GaGu^{180}$** | **-0.1992** | |
| | | | | $WAIC_{best} - WAIC_{true}$: | **-0.0094** | |
| $Cl^0$ $Cl^{90}$ | $Gu^{180}$ | -0.0253 | 62 m | Ga | -0.0079 | 5 m |
| | $Gu^{180}Cl^{90}$ | -0.2383 | | $InGaGu^{180}Gu^{270}Gu^0Gu^{90}$ | -0.1904 | |
| | $Gu^{180}Cl^{90}Cl^0$ | -0.2506 | | $InGaCl^0Cl^{90}Cl^{180}Cl^{270}$ | -0.2330 | |
| | $Gu^{180}Cl^{90}Cl^0In$ | -0.2509 | | **$Cl^0Cl^{90}$** | **-0.2361** | |
| | $Gu^{180}Cl^{90}Cl^0InFr$ | -0.2508 | | | | |
| | **$Cl^0Cl^{90}$** | **-0.2586** | | | | |
| $Cl^{180}$ $Gu^{270}$ | $Gu^{270}$ | -0.0242 | 69 m | Ga | -0.0083 | 6 m |
| | $Gu^{270}Cl^{180}$ | -0.2499 | | $InGaGu^{180}Gu^{270}Gu^0Gu^{90}$ | -0.2277 | |
| | $Gu^{270}Cl^{180}Gu^{180}$ | -0.2517 | | $InGaCl^0Cl^{90}Cl^{180}Cl^{270}$ | -0.2535 | |
| | $Gu^{270}Cl^{180}Gu^{180}In$ | -0.2518 | | **$GaCl^{90}Cl^{180}$** | **-0.2549** | |
| | $Gu^{270}Cl^{180}Gu^{180}InCl^0$ | -0.2518 | | | | |
| | $Gu^{270}Cl^{180}Gu^{180}InCl^0Fr$ | -0.2518 | | | | |
| | **$Cl^{180}Gu^{270}$** | **-0.2500** | | | | |
| | | | | $WAIC_{best} - WAIC_{true}$: | **-0.0098** | |

Table A6: The model selection histories for 3-element mixtures with constant $\theta$ and variable $\phi$

| True Model | Greedy | | | Heuristic | | |
|---|---|---|---|---|---|---|
| | Search attempts | WAIC | T | Search attempts | WAIC | T |
| Ga $Cl^{90}$ $Gu^0$ | $Gu^0$ | -0.1399 | 44 m | Ga | -0.1252 | 6 m |
| | $Gu^0Cl^{90}$ | -0.2494 | | $InGaGu^{180}Gu^{270}Gu^0Gu^{90}$ | -0.2481 | |
| | $Gu^0Cl^{90}Cl^0$ | -0.2519 | | $InGaCl^0Cl^{90}Cl^{180}Cl^{270}$ | -0.2565 | |
| | $Gu^0Cl^{90}Cl^0Fr$ | -0.2518 | | **$GaCl^{90}Cl^{180}$** | **-0.2564** | |
| | **$Cl^{90}Gu^0$** | **-0.2494** | | | | |
| | $WAIC_{best} - WAIC_{true}$: | **-0.0036** | | $WAIC_{best} - WAIC_{true}$: | **0.0014** | |
| Fr $Cl^{90}$ $Gu^0$ | Fr | -0.0591 | 77 m | Ga | -0.0489 | 6 m |
| | $FrCl^{90}$ | -0.1460 | | $InGaGu^{180}Gu^{270}Gu^0Gu^{90}$ | -0.1573 | |
| | $FrCl^{90}Gu^0$ | -0.1730 | | $InGaCl^0Cl^{90}Cl^{180}Cl^{270}$ | -0.1578 | |
| | $FrCl^{90}Gu^0Cl^{180}$ | -0.1736 | | **$GaCl^{90}Cl^{180}$** | **-0.1621** | |
| | $FrCl^{90}Gu^0Cl^{180}In$ | -0.1734 | | | | |
| | **$Gu^0Cl^{90}Fr$** | **-0.1731** | | | | |
| | | | | $WAIC_{best} - WAIC_{true}$: | **0.0059** | |
| Fr $Cl^{180}$ $Gu^{270}$ | Fr | -0.0741 | 87 m | Ga | -0.0618 | 9 m |
| | $FrCl^{180}$ | -0.1513 | | $InGaGu^{180}Gu^{270}Gu^0Gu^{90}$ | -0.1567 | |
| | $FrCl^{180}Gu^{270}$ | -0.1707 | | $InGaCl^0Cl^{90}Cl^{180}Cl^{270}$ | -0.1670 | |
| | $FrCl^{180}Gu^{270}Cl^{90}$ | -0.1708 | | $InGaGu^{270}Cl^{180}Cl^{270}$ | -0.1680 | |
| | $FrCl^{180}Gu^{270}Cl^{90}Gu^{180}$ | -0.1711 | | $InGaGu^{270}Cl^{180}Gu^{90}$ | -0.1695 | |
| | $FrCl^{180}Gu^{270}Cl^{90}Gu^{180}$-$Cl^0$ | -0.1710 | | **$InGu^{270}Cl^{180}$** | **-0.1735** | |
| | **$Gu^{270}Cl^{180}Fr$** | **-0.1703** | | | | |
| | | | | $WAIC_{best} - WAIC_{true}$: | **-0.0011** | |
| $Gu^0$ $Gu^{180}$ $Cl^{90}$ | $Gu^0$ | -0.1695 | 47 m | Ga | -0.1477 | 11 m |
| | $Gu^0Cl^{90}$ | -0.3040 | | $InGaGu^{180}Gu^{270}Gu^0Gu^{90}$ | -0.2986 | |
| | $Gu^0Cl^{90}Gu^{180}$ | -0.3234 | | $InGaCl^0Cl^{90}Cl^{180}Cl^{270}$ | -0.3033 | |
| | $Gu^0Cl^{90}Gu^{180}Cl^{180}$ | -0.3233 | | $GaGu^{180}Cl^{90}Cl^{180}$ | -0.3054 | |
| | **$Gu^{180}Cl^{90}Gu^0$** | **-0.3234** | | $GaGu^{180}Cl^{90}Gu^0$ | -0.3111 | |
| | | | | **$Gu^{180}Cl^{90}Gu^0$** | **-0.3113** | |

Table A7: The model selection histories for 3-element mixtures with constant $\phi$ and variable $\theta$

| True Model | Greedy Search attempts | WAIC | T | Heuristic Search attempts | WAIC | T |
|---|---|---|---|---|---|---|
| $Ga$ $Cl^{90}$ $Gu^0$ | $Fr$ | -0.0177 | 66 m | $Ga$ | -0.0142 | 13 m |
| | $FrGu^{270}$ | -0.1284 | | $InGaGu^{180}Gu^{270}Gu^0Gu^{90}$ | -0.1291 | |
| | $FrGu^{270}Gu^0$ | -0.1407 | | $InGaCl^0Cl^{90}Cl^{180}Cl^{270}$ | -0.1289 | |
| | $FrGu^{270}Gu^0Cl^0$ | -0.1423 | | $InCl^0Gu^{270}Gu^0$ | -0.1317 | |
| | $FrGu^{270}Gu^0Cl^0Cl^{180}$ | -0.1435 | | $InCl^0Cl^{90}Gu^0$ | -0.1346 | |
| | $FrGu^{270}Gu^0Cl^0Cl^{180}In$ | -0.1432 | | $InCl^0Cl^{90}Cl^{180}$ | -0.1301 | |
| | **$Gu^0Gu^{270}$** | **-0.1451** | | **$GaCl^{90}Cl^{180}$** | **-0.1313** | |
| | $WAIC_{best} - WAIC_{true}$: | **0.0132** | | $WAIC_{best} - WAIC_{true}$: | **0.0068** | |
| $Fr$ $Cl^{90}$ $Gu^0$ | $Fr$ | -0.0265 | 71 m | $Ga$ | -0.0192 | 9 m |
| | $FrGu^{270}$ | -0.1290 | | $InGaGu^{180}Gu^{270}Gu^0Gu^{90}$ | -0.1411 | |
| | $FrGu^{270}Gu^0$ | -0.1445 | | $InGaCl^0Cl^{90}Cl^{180}Cl^{270}$ | -0.1429 | |
| | $FrGu^{270}Gu^0Cl^{180}$ | -0.1450 | | $InGaGu^{180}Cl^{90}Cl^{180}$ | -0.1474 | |
| | $FrGu^{270}Gu^0Cl^{180}Cl^0$ | -0.1466 | | $InGaGu^{180}Cl^{90}Gu^0$ | -0.1472 | |
| | $FrGu^{270}Gu^0Cl^{180}Cl^0In$ | -0.1468 | | **$InCl^{90}Gu^0$** | **-0.1477** | |
| | **$Gu^0Gu^{270}$** | **-0.1451** | | | | |
| | $WAIC_{best} - WAIC_{true}$: | **0.0109** | | $WAIC_{best} - WAIC_{true}$: | **0.0010** | |
| $Fr$ $Cl^{180}$ $Gu^{270}$ | $Fr$ | -0.0129 | 61 m | $Ga$ | -0.0185 | 6 m |
| | $FrGu^{270}$ | -0.1105 | | $InGaGu^{180}Gu^{270}Gu^0Gu^{90}$ | -0.1309 | |
| | $FrGu^{270}Gu^0$ | -0.1237 | | $InGaCl^0Cl^{90}Cl^{180}Cl^{270}$ | -0.1326 | |
| | $FrGu^{270}Gu^0Cl^{180}$ | -0.1254 | | $InGu^{270}Cl^{180}$ | -0.1393 | |
| | $FrGu^{270}Gu^0Cl^{180}Gu^{180}$ | -0.1248 | | $InGu^{270}Gu^0$ | -0.1334 | |
| | **$Gu^0Gu^{270}Fr$** | **-0.1234** | | **$InGu^{270}Gu^0$** | **-0.1326** | |
| | $WAIC_{best} - WAIC_{true}$: | **0.0094** | | $WAIC_{best} - WAIC_{true}$: | **0.0088** | |
| $Gu^0$ $Gu^{180}$ $Cl^{90}$ | $Gu^0$ | -0.0756 | 55 m | $Ga$ | -0.0454 | 7 m |
| | $Gu^0Cl^{90}$ | -0.2380 | | $InGaGu^{180}Gu^{270}Gu^0Gu^{90}$ | -0.2476 | |
| | $Gu^0Cl^{90}Cl^0$ | -0.2556 | | $InGaCl^0Cl^{90}Cl^{180}Cl^{270}$ | -0.2459 | |
| | $Gu^0Cl^{90}Cl^0Ga$ | -0.2591 | | $GaCl^0Gu^{270}Gu^0$ | -0.2493 | |
| | $Gu^0Cl^{90}Cl^0GaCl^{270}$ | -0.2590 | | $GaCl^0Cl^{90}Gu^0$ | -0.2559 | |
| | **$Cl^0Cl^{90}Gu^0$** | **-0.2555** | | **$Cl^0Cl^{90}Gu^0$** | **-0.2538** | |
| | $WAIC_{best} - WAIC_{true}$: | **0.0026** | | $WAIC_{best} - WAIC_{true}$: | **0.0006** | |

