# OpenReview forum: "Parametric Copula-GP model for analyzing multidimensional neuronal and behavioral relationships"
_ICLR.cc/2021/Conference — Reject_

### Official Review · AnonReviewer2 · 2020-10-28
**Promising entropy estimation on real data but some fuzzy details**

**Rating:** 7
**Confidence:** 4

**Review:**

This manuscript models the conditional joint distribution over variables by using Copula models and copula vines.  The experimental data shows that when the observed variables are highly correlated that the proposed approach improves estimation of entropy over competing benchmark approaches (MINE and KSG) when the variables are highly correlated. Synthetic results demonstrate good improvements, and application to real scientific data seems promising.

Strengths:
Application to real, novel scientific data shows potential utility of the model.
Synthetic results show a good improvement over competing methods, albeit in a limited setup (highly correlated variables)
Mixtures of copulas seems an effective way to produce model complexity, and by linking it to a GP can make sure that it's smooth over the conditioned variable x.

Weaknesses:
Not all modeling steps are clear.  In particular, the interaction between the GP prior and the model selection step is undescribed.  Since there are multiple ways to do this, needs a full description.
No comparison to more neuroscience focuses techniques.
Experimental setup and utility is not fully described.
Lacks ablation studies to elicit key model components.

Questions:
The choice to make the link function on the different copula families all dependent on the same $f$ seems like a strange and limiting choice.  Why was this choice made?

The choice to model calcium trace level and not neural spiking seems mathematically convenient for this method, but it's not clear to me that this is the correct scientific choice.  Typically, calcium imaging traces are preprocesses to extract spikes.  Is there a scientific rationale for using the raw data?  Or was this primarily motivated by avoiding discrete measurements that would be harder to model in the copula?

Please describe more of the scientific setup.  For example, why is there any relationship to licks outside of the reward?

What is the scientific question on this neuroscience application?  Why is entropy relevant, rather than using one of the many predictive problems?

Update after author response:
Most of my methodological concerns have been address (except for the ablation studies).  The scientific application here is not super well-motivated.  It would improve the article greatly to show a greater utility (or at least, clearly describing future utility for answering scientific questions).  It is mentioned that "A full application of the method to study the dependence of contextual signals in mouse visual cortex will be the focus of a follow-up publication."  That's vague; it would be nice to at least clearly discuss how this could be used to facilitate or enhance these scientific experiments.

---

> ### Author Response · Authors · 2020-11-17
> **Part 1: authors clarify modelling procedure**
>
> The authors would like to thank AnonReviewer2 for helpful feedback. Below we clarify the details of the method and the experimental setup (in a second comment).
>
> _>Not all modeling steps are clear. In particular, the interaction between the GP prior and the model selection step is undescribed. Since there are multiple ways to do this, needs a full description._
> We provide the information on the GP priors and model selection in Appx. A.
> We also intend to release the code together with the paper upon its acceptance. We now share it with the reviewers and AC (see a separate post).
>
> _>No comparison to more neuroscience focuses techniques._
> We would be grateful if the reviewer could provide more details on this comment, e.g. which particular techniques would be interesting for the comparison from their perspective.
>
> We can comment on the key differences between our Copula-GP model and some of the methods, typically used in neuroscience literature for the analysis of population recordings. First, unlike linear methods (such as PCA, SVD, CCA...), copula mixtures describe non-linear statistical relationships between variables. If our model was restricted to the use of only Gaussian copulas, it would have been linear and would be equivalent to the typical noise correlation analysis. Second, unlike other statistical non-copula approaches (such as GLM or GPFA), our Copula-GP framework allows us to model the dependencies between elements with utterly different statistics (e.g. licks vs. velocity). In addition, Copula-GP explicitly represents the dependencies as a function of position, revealing insightful information about the task structure. To the best of our knowledge, there are currently no other methods with this combination of features. We have added the above explanation to the discussion.
>
> _>Lacks ablation studies to elicit key model components._
> Copula-GP is a Bayesian model. A big advantage of Bayesian models is that the role of each of the components (copula elements) can be studied without the need for ablation. For example, plots in Fig.A7 (right) show priors over model parameters, where the mixing parameters (rightmost) illustrate the key components that describe the data in different contexts (i.e. in different places in VR in our case).
> Additionally, we explain how to attribute the data-points to certain components of the models in Appx. B.3.
>
> _>The choice to make the link function on the different copula families all dependent on the same f seems like a strange and limiting choice. Why was this choice made?_
> Our copula mixture models are parameterized with 2K-1 Gaussian processes ($f$s), so each copula element depends on its own $f$ (K independent GPs in total) + the mixture of copulas is parameterized by (K-1) additional mixing parameters.
> This is the most general parametrization, which can then be constrained. For example, for performing factor analysis, one may consider [linearly dependent GPs](https://docs.gpytorch.ai/en/latest/variational.html#lmcvariationalstrategy).
>
> _>The choice to model calcium trace level and not neural spiking seems mathematically convenient for this method, but it's not clear to me that this is the correct scientific choice. Typically, calcium imaging traces are preprocesses to extract spikes. Is there a scientific rationale for using the raw data? Or was this primarily motivated by avoiding discrete measurements that would be harder to model in the copula?_
> The rationale for using the raw data (instead of average firing rates, for example, which are also continuous) was motivated by avoiding additional steps in data processing that introduce biases through arbitrary parameters.  Deconvolution algorithms require fine-tuning of parameters that depend on the type of calcium indicator used (e.g. decay time of the fluorescence signal) and the cell type under investigation (e.g. spiking rate).
> Indeed, applying copula analysis to continuous variables is easier than to discrete ones. Yet, the latter is also possible and we plan to extend our package to discrete data as well (using methods from [Onken, 2016](https://papers.nips.cc/paper/6069-mixed-vine-copulas-as-joint-models-of-spike-counts-and-local-field-potentials)).

---

> > ### Author Response · Authors · 2020-11-17
> > **Part 2: experimental setup**
> >
> > _>Experimental setup and utility is not fully described. … Please describe more of the scientific setup. For example, why is there any relationship to licks outside of the reward?_
> > The animals were able to lick anytime along the virtual track. Before learning the task, the mice were licking in many locations outside the reward zone. Even after learning, some licks were still made in non-rewarded locations. All licks were recorded such that we could correlate the lick positions and the delivery of the reward (only in the reward zone). The data in Fig.3 corresponds to the first experimental day, when the mice were learning the task and as a consequence, their task performance was low with many licks outside the reward zone. Mice also tend to reduce running speed when they lick, and speed is known to strongly modulate neuronal activity in V1. Our model allows us to study the joint relationship between the licks, velocity and neuronal activity. However, interpreting such a model and disentangling different factors requires a few more modelling steps and likely additional control experiments that lay beyond the scope of this paper and that we hope to cover in subsequent publications. We have clarified the goals and the scope of the study in the introduction (last paragraph).
> >
> > _>What is the scientific question on this neuroscience application?_
> > The neuroscience data in this paper serve as a source of natural statistics for validation of our newly developed modelling framework. We were analysing what kind of dependencies are present in neuronal and behavioural data (see percentages in Fig.1 and density plots in Fig. 3C-D) and testing the ability of our model to approximate them (see goodness-of-fit) and scale to the large neuronal population recordings (Fig. 3H). Before using this method to address a scientific question, we wanted to validate the method on neuronal data. A full application of the method to study the dependence of contextual signals in mouse visual cortex will be the focus of a follow-up publication.
> >
> > _>Why is entropy relevant, rather than using one of the many predictive problems?_
> > As was correctly noted by AnonReviewer4, our Copula-GP method is not limited to entropy estimation, but provides a model of the probability distribution of the neuronal and behavioral variables given the position. Therefore, depending on the particular scientific question, we can calculate any expectation under this distribution using this model. We can also use Copula-GP for Bayesian decoding, which is one of the many predictive approaches mentioned by the reviewer. We encourage the reviewer to explain the meaning of ‘predictive problems’ in this context, if we have not answered the raised question.
> >
> > In this paper, we chose to focus on mutual information as the most general of all measures, applicable to a wide range of applications. It is also perfect for evaluating the performance of the method and validating its implementation in our newly developed python package. It also gives an upper bound to the amount of information that can be extracted by a decoding algorithm [Quiroga, 2009](https://www.nature.com/articles/nrn2578#Sec4).
> >
> > In our application in particular, the conditional entropy is showing the variability of the data at different locations. We mention that part of this variability (the first peak) can be explained by the animal's behaviour (velocity) and the other part by the task context (reward zone) (see the second to last paragraph of the results). As mentioned previously,  a detailed analysis of the dependence of contextual signals in mouse visual cortex will be the focus of a follow-up publication.

---

> > ### Comment · AnonReviewer2 · 2020-11-18
> > **Reasonable responses; still recommend ablation studies.**
> >
> > This largely seems reasonable, and the additional details have enhanced the manuscript.
> >
> > Re: "Our copula mixture models are parameterized with 2K-1 Gaussian processes (s), so each copula element depends on its own  (K independent GPs in total) + the mixture of copulas is parameterized by (K-1) additional mixing parameters."  That makes way more sense; however, you should revise that section for clarity, because that is not how I read the section, nor is it when I reread it.  I would add a comment under bivariate copula families because the notation is overloaded.
> >
> > Regarding "A big advantage of Bayesian models is that the role of each of the components (copula elements) can be studied without the need for ablation."  I strongly disagree.  You definitely can post-hoc analyze these components, but there are strong effects of the prior and it's unclear which part of the model is necessary.  It is standard in the Bayesian modeling literature to show improvements in predictions or estimations as you add or remove parts of the model (e.g., a hierarchical structure improves over an independent structure).

---

> > > ### Author Response · Authors · 2020-11-19
> > > **Added ablation, but for our mixtures the result is trivial**
> > >
> > > Indeed, without ablation, we can only analyze the key components *for the selected solution*.
> > >
> > > However, the components of the copula mixture are qualitatively different, so we would expect that the result of the ablation study would be trivial in most of the cases. From our identifiability studies, we know that the only case where 1 element can be mistaken for 2 other elements is a Gaussian copula vs. Clayton & Gumbel (in the mixtures with 3+ components). Essentially, we perform a mini-ablation study at the end of our heuristic algorithm (see lines 20-30 in Algorithm 2), if it finds some combinations which can in fact be substituted by a single Gaussian copula.
> > >
> > > Since we propose our framework as a general tool that can be potentially used with different copula elements, more situations that require ablation may arise. Therefore, we have added ablation into our package and performed it on the models from Fig.3C-D (same data as Fig.A7) [see notebooks/Ablation.ipynb]. All models after ablation had higher WAIC than the originally selected model. Ablation of the Clayton copula in Fig.3D increased WAIC the most (from -0.043 to -0.033), while both elements in Fig.3C were equally important (WAIC after ablation was the same (within tolerance)). We have added the above description to Appx. C.3.
> > >
> > > The authors are grateful to the Reviewer for suggested improvements to the package and the manuscript.
> > >
> > > UPD: we have also revised the section about 2K-1 Gaussian processes

---

### Official Review · AnonReviewer4 · 2020-10-28
**Cool method, poor motivation.**

**Rating:** 5
**Confidence:** 4

**Review:**

The authors develop a Gaussian process vine copula model, very much in the flavor of the modeling approach of Lopez-Paz et al. (2013). The improvement to the earlier work seems to be a framework for flexible copula modeling including a copula mixture model, approximate inference, model selection, and calculation of mutual information. The paper on its own is a modest improvement on existing work and is both an engineering accomplishment and has the potential for a useful model. The paper is exceptionally well-written and clear. I found it a breeze to read and I credit the authors for that. However, both the validation and the motivation for the model (namely characterizing the probabilistic relationships for neural and behavioral variables) seems particularly thin and could be substantially improved. The paper falls short on these points to such a degree that I am hesitant to recommend acceptance since it notably impacts my evaluation of the significance of the works

## Major points:

First, the authors claim (page 7, paragraph 2) that their model out-performs all non-parametric methods. This is by no means obvious form Fig2b,c. Moreover, If the model performs "similarly" to MINE, then it is maybe not worth using when a convenient technique already exists. What is the advantage of the present model?

Second, the authors show that the estimation of mutual information is only unbiased for a narrow range  of distributions (Gaussian or transformed Gaussian for small dimensions according to Fig 2) and fails for the heavy-tailed Student's-T. However, many neural and behavioral variables are themselves heavy-tailed and the authors did not demonstrate that the real data are sub-gaussian.

Third, it seems like there was a wasted opportunity with this paper. The authors spent $\approx$ 2/3 of a page discussing the estimation of mutual information without motivating why that was a good example metric that could be derived from their model. This modeling framework is an opportunity to determine virtually any expectation over the entire distribution and it is entirely possible that MI is neither all that interesting, nor does it play to the strengths of the model. They then describe changes  to the pairwise distributions of variables from the copula model but we didn't need the copula model to estimate.

Finally, what are we to make of the real data results *vis-a-vis* the validation experiments with simulated data in Fig 2? Besides what I mentioned above regarding the tails of the real data, its not clear that the variance explained wouldn't behave differently. Could the authors report the variance explained for their simulation experiments?

## Minor points:
The authors state (page 7, last paragraph) That the "stimulus-related changes in the joint variability of the two neuronal signals are commonly described as _noise correlations_." But, isn't that the definition of _signal correlations_?

the authors state (page 8, paragraph 5)
>The Copula-GP “estimated” (dashed line) almost perfectly matches the “integrated” result, which suggests that the model was able to tightly approximate both $p(u^x|x)$ and $p(u^x)$, and, as a result, $I(x, \{u|^x_{i<N} \})$.

However, it is not clear at all that the later statement regarding $I(x, \{u|^x_{i<N} \})$ follows from the former. In fact Figure 2 demonstrates that the integrated result may not estimate $I(x, \{u|^x_{i<N} \})$ well at all.

---

> ### Author Response · Authors · 2020-11-14
> **Part 1: Figures 2b-c are focussed on limitation of MI estimators, while Copula-GP is guaranteed to produce unbiased estimates when the true dependency matches the parametric model.**
>
> We appreciate the thoroughness of the review and the recommendations for improving the motivation of the study.
>
> _>First, the authors claim (page 7, paragraph 2) that their model out-performs all non-parametric methods. This is by no means obvious form Fig2b,c._
> We are grateful to AnonReviewer4 for explaining the misleading impression created by Fig. 2. This figure is focused on the specifically difficult cases for the information estimators. Panel (A) shows a trivial example, in which MC integration of our model is guaranteed to produce an unbiased entropy estimate, assuming that the implementation of log-likelihood and sampling is correct, which is ensured by the tests (see the code, provided to reviewers in a private comment) and correctly estimated parameters. The bias in parameter estimation is addressed separately in validation of parameter inference (see Appx. B and integration tests in the code). Yet, even such a trivial example becomes challenging for the non-parametric models in higher dimensions. Note, that the estimate is unbiased not only for Gaussian, but also for Clayton, Frank or Gumbel copula, or any linear mixture of those.
>
> Next, the example in panel (B) is specifically designed as a challenging case for the MINE estimator, while the one in panel (C) is extremely difficult for our estimator. Thus, the aim of Fig. 2 was to show the limitations of these methods and differences between them, rather than their typical performance (which would depend on the application).
>
> We have added a few introductory sentences before Fig. 2, explaining that the parametric model is guaranteed to produce unbiased estimates, if the true underlying dependence matches the parametric model. To eliminate any concerns regarding unbiased estimates of the heavy-tailed distributions, we have added an analog of Fig.2A with a Clayton copula into Appx. B.3.
> We should also emphasize, that MINE estimates depend on the choice of hyper-parameters.  In fig. 2B, MINE with 200 and 500 hidden units overestimated the mutual information while being theoretically a lower bound. We now change the line style in Fig.2 to draw the reader's attention to this fact, which was previously only explained in the text.
>
> _>Moreover, If the model performs "similarly" to MINE, then it is maybe not worth using when a convenient technique already exists. What is the advantage of the present model?_
> Our model is technically a semi-parametric model, while MINE is non-parametric. We would like to show that Copula-GP can take the best of both worlds with the use of the mixture models. It has a small number of elements and explicit parameterization, so it remains quite interpretable. The interpretable parameters include the copula parameters of the heavy-tailed copula families and the family identities (see Appx. C.2 on model interpretation). Yet, what is impressive, is that Copula-GP performs *at least* similarly to MINE on the examples where it is not guaranteed to perform well. Such a good performance is made possible by mixing of copulas.
> We acknowledge that we have not properly conveyed this message and we now added an explanation to the Discussion (see the sentence starting with ‘Unlike black-box ...’).
>
> _>Second, the authors show that the estimation of mutual information is only unbiased for a narrow range of distributions ... and fails for the heavy-tailed Student's-T._
> The MC estimate is unbiased for any such continuous distribution, for which the dependence can be represented as a linear mixture of the copula families used in our model. Student-T distribution can not be represented as a linear mixture, only approximated. Therefore, Copula-GP approximates Student-T with a mixture of Gaussian and Gumbel copulas, and estimates the mutual information. The result appears to be closer to the ground truth than the estimates from non-parametric methods (fig. 2B, non-dashed lines). We also provide variance explained (R^2) for the approximation of the Student T distribution (99% in bivariate case), following one of the comments below.
> Note that the example in Fig.2B is challenging mainly due to the chosen parametrization of Student-T copula: constant correlation, variable degrees of freedom.

---

> ### Author Response · Authors · 2020-11-14
> **Part 2: Experimental data is heavy tailed and Copula-GP models that explicitly**
>
> _>However, many neural and behavioral variables are themselves heavy-tailed and the authors did not demonstrate that the real data are sub-gaussian._
> Yes, the experimental data we analyzed is heavy-tailed (see Figure 3C-D & Figure A7 (former A5)), and our model represents these heavy tails with the Clayton or Gumbel distributions. In Figure 1, we show how often each of the copula families was used in the mixtures, which were selected by our model selection algorithm for modeling neuronal data. One can see that 57% of copula elements were Clayton and only 10% were Gaussian, suggesting that heavy-tailed bivariate dependencies were very common.
>
> _>Third, it seems like there was a wasted opportunity with this paper..._
> We would like to thank AnonReviewer4 for pointing out that the motivation was not well explained. We focus on mutual information for 2 reasons: 1) it is very general and allows us to compare semi-parametric and non-parametric models in one benchmark; 2) it is relevant for a wide range of neuroscience applications. It also covers both sampling and log-likelihood estimation, and thereby serves as a perfect validation metric for our newly developed GPU-accelerated copula package.
>
> The alternative applications of the parametric copula models are indeed numerous. We mention them briefly on page 8, yet the details on their application lay beyond the scope of this paper. They are also well described in other papers, which are focussed specifically on copulas, and hence are not novel. Although, following this and other reviewer’s comments (AnonReviewer1), we added one method for outlier detection, that is specific for our linear model (see Appx.C.3).
>
> _>Finally, what are we to make of the real data results vis-a-vis the validation experiments with simulated data in Fig 2?_
> The validation experiments with simulated data lead us to the information estimate shown in the final figure: Figure 3H. We do not have the ground truth for the neuronal data, but we propose 2 methods for information estimation and they agree very well. At the same time, MINE and KSG provide much lower estimates on the large subsets of data.
>
> _>Besides what I mentioned above regarding the tails of the real data, its not clear that the variance explained wouldn't behave differently. Could the authors report the variance explained for their simulation experiments?_
> Yes. Using the exact same procedure as was applied to the data in Fig. 3, we calculated the R^2 for the bivariate (2D) models for Student T distribution and transformed Gaussian distribution. The result shows that Copula-GP explains 99% and 98% respectively. The code is provided in this notebook: <link-provided-to-reviewers-in-a-private-comment>/notebooks/vine/VarExp4SimData.ipynb

---

> ### Author Response · Authors · 2020-11-14
> **Part 3: Reply to minor comments**
>
> _>The authors state (page 7, last paragraph) That the "stimulus-related changes in the joint variability of the two neuronal signals are commonly described as noise correlations." But, isn't that the definition of signal correlations?_
> The term ‘noise correlations’ often appears confusing, but it is used correctly in our statement. Signal correlations by definition are the correlations of the mean response of each single neuron with the stimulus. In our model, marginals account for signal correlations, while copula describes the noise correlations. This separation became possible due to the use of conditional marginals, instead of unconditional, which is one of the key innovations of our model.
> The definitions are well illustrated, for example, in Bejjanki, Vikranth R., et al. ["Noise correlations in the human brain and their impact on pattern classification."](https://journals.plos.org/ploscompbiol/article?id=10.1371/journal.pcbi.1005674) PLoS computational biology 13.8 (2017): e1005674.
>
> _>the authors state (page 8, paragraph 5) The Copula-GP “estimated” (dashed line) almost perfectly matches the “integrated” result, which suggests that the model was able to tightly approximate both p(ux|x) and  p(ux), and, as a result, I(x,u|i<Nx). However, it is not clear at all that the later statement regarding I(x,u|i<Nx) follows from the former._
> It follows from the former via eq.3. We added a reference to that equation after the statement.
>
> _>In fact Figure 2 demonstrates that the integrated result may not estimate I(x,u|i<Nx) well at all._
> That happens due to model mismatch. As we mentioned earlier, the examples in Fig 2 are quite challenging for all estimators. The goal was to push all of the estimators to the extreme, while we expect better performance in real life applications (as it is known that elliptic copulas typically approximate real life dependencies quite well, and we confirm it for the neuronal data with the goodness-of-fit measure).

---

### Official Review · AnonReviewer1 · 2020-10-30
**Very interesting application but novelty is below the ICLR level**

**Rating:** 5
**Confidence:** 4

**Review:**

- This is an interesting neuroscience application where Copula estimation has shown to be effective.

- Having said that, I believe that the technical novelty is minimal. To the authors' credit, they did not claim a huge theoretical edge. They honestly reported the findings of the paper.

- The paper is well written. Ideas flow very neatly.

- p2: "It was previously shown that such a combination of parametric copula models with GP priors ...": Precisely.. Therefore, given the work by Hernandez-Lobato et al. (2013), as well as the works which have eventually built on top of it, novelty of the proposed method is minimal. I am not saying the application is not useful though.

- Along the same lines from above, equations (2), (3) and (4) which depict the core of the method, are all taken from seminal previous works of copula mixture estimation.

- p3: "Since none of the aforementioned families alone could describe such conditional dependency, we combined multiple copulas into a linear mixture model": Is it possible to elaborate a little bit more on whether this is actually the best technical choice here? For instance are there any side effects (e.g. computational) resulting from using this linear mixture?

---

> ### Author Response · Authors · 2020-11-14
> **Novelty of Copula-GP (conditional marginals & mixture models) and distinction from previous works**
>
> We would like to thank AnonReviewer1 for the constructive comments, which pointed out that the difference of our model from the Hernandez-Lobato et al. 2013 was not explained very well and novelty was not emphasized. Therefore, we would like to start the discussion by explaining this difference (and by modifying the paper accordingly).
>
> First, and a major difference, is that the marginals are *conditional*. This detail dramatically changes the dependence that the copula is supposed to model (see $\mathbf{u}$ vs. $\mathbf{u}^x$ throughout the paper). The difference can be best illustrated with a toy example, which we now provide in Appx.C.1.
> In brief, we consider a pair of simulated neurons, which both fire strongly and independently in a certain context (for certain values of x). We apply unconditional (like in Hernandez-Lobato et al. paper) and conditional (like Copula-GP) marginal models to the virtual recordings from these simulated neurons. We show that empirical copula marginals in an unconditional model are far from being uniform. Therefore, one of the modeling assumptions does not hold, and the copula model fit is poor. This illustrates that copula models with unconditional marginals are not applicable to the complex neuronal data with highly variable marginal statistics. We also show that the model with conditional marginals is more interpretable. In the toy example, our Copula-GP model correctly shows no noise correlation between simulated neurons (see Fig. A6).
>
> The second difference is the use of mixture models. We observed that the same pair of neurons may exhibit, for example, an upper tail dependence in one zone of the VR environment, and lower tail dependence in the other (see Figure 3D and Figure A7B (former A5)), or have a heavy tail in only one particular zone (Figure 3C and Figure A7A (former A5)). Therefore, a linear mixture of copula models with different tail dependencies was a straightforward solution.
>
> Finally, there are technical improvements over the Hernandez-Lobato 2013 paper. We use scalable SVI instead of EP and GPU-accelerated PyTorch and GPyTorch libraries. To the best of our knowledge, there are currently no other libraries that implement copula distributions as pytorch distributions. We believe that this makes our code significantly more reusable. The modular structure of our package allows us to change the copula elements or the model selection algorithms with ease. Due to the sequential training of the vines, it is also easy to cast the training onto a multi-GPU system (or even simulate heterogeneously on CPUs and GPUs). We are happy to share the code privately with the reviewers and AC (see a separate private comment), and we intend to make it publicly available upon the acceptance of the paper.
>
> Also, see Appx. B.4 (former B.3) for the direct comparison with the Hernandez-Lobato model on the UCL Shuttle dataset. It shows that our Copula-GP model produces higher log-likelihood on the test data, compared to the original Hernandez-Lobato model.
>
> _>Is it possible to elaborate a little bit more on whether this is actually the best technical choice here? For instance are there any side effects (e.g. computational) resulting from using this linear mixture?_
> We now provide a detailed discussion of the advantages of using linear mixtures (Appx.C.2&4). First, linear mixtures are quite interpretable. We can look at the mixing coefficients for the copula elements and immediately tell the qualitative properties of the dependence (see Appendix C.2 with two examples). Second, we can highlight and backtrack the data-points that belong to certain heavy tails. We briefly mention this on page 8, but the details lay beyond the scope of this paper. Following this reviewer’s comment, we now added one method for heavy tail detection (Appx.C.4), which is specific for our mixture model.
>
> _>equations (2), (3) and (4) which depict the core of the method, are all taken from seminal previous works of copula mixture estimation._
> While equations (2) and (4) are widely known and are provided as a necessary background on information measures and copulas, we are not aware of any other studies mentioning anything like equation (3) and the associated methodological proposal. The equation (3) is proposed by us for our Copula-GP model with *conditional* marginals, and we are not aware of any other works with such marginal models.

---

### Official Review · AnonReviewer3 · 2020-10-30
**application of copula mixtures to model time-varying multimodal data**

**Rating:** 6
**Confidence:** 3

**Review:**

The authors exploit the expressive power of Copula mixtures to model time-varying multi-modal data, and employ Gaussian Processes to model the time-varying copula parameters. They demonstrate the efficacy of their method using information theoretic metrics on a synthetic dataset and a real-world joint neural-behavioral dataset from a neuroscience experiment.  Results demonstrate that the proposed techniques are comparable to the state of the art nonparametric methods, while being more scalable due to the use of stochastic optimization based methods that are commonly used with parametric methods.

The paper was quite thorough in the motivation, development and empirical analysis of the proposed technique. The use of Copulas, that are commonly employed in the Finance community, but are rare in statistical neuroscience, should interest the more theoretically inclined reader. Given the accelerating trend of collecting long-term joint neural and behavior in experimental neuroscience, the authors make an interesting and timely contribution to the statistical neuroscience literature.

I found that the motivations of the paper were difficult to extract from the Introduction, as there was substantial use of jargon and the intuition behind the technical results were not accessible. To encourage the adoption of their methods in the neuroscience community, the authors should consider improving the readability of their manuscript by making it more friendly to the non-statistician reader.

---

> ### Author Response · Authors · 2020-11-17
> **A friendlier motivation for our study**
>
> We are grateful to AnonReviewer3 for the feedback on our paper.
>
> _>I found that the motivations of the paper were difficult to extract from the Introduction, as there was substantial use of jargon and the intuition behind the technical results were not accessible._
>
> We have now clarified the motivations of our article and extended the introduction to include a more accessible description of the key theoretical concepts (e.g. mutual information and marginal statistics). We hope that these changes would improve the readability of our paper.

---

### Decision · Program_Chairs · 2021-01-07
**Final Decision**

**Decision:**

Reject

**Comment:**

Summary: The authors built on existing work of GP vine copula
models. Some modifications are made, to conditional marginals and
mixing. Applications to mutual information estimation are discussed
and evaluated, and the approach is applied to joint
neural/behavioral data.


Discussion:
Strengths mentioned in the reviews are that the
application is (from a neuroscience perspective) interesting, that
estimating mutual information is an important problem, and that the
paper is very well-written. Weaknesses are the limited novelty (from a
machine learning perspective), and weak empirical validation.

The authors have responded in detail, and were able to clarify a
number of unclear points. Clearly, however, the main criticisms noted
above are hard to address in discussion.

Despite the paper being overall clearly written, I agree with
reviewers that it is hard to tell from
abstract and introduction where the paper is going (even after
modifications made by the authors in the course of the discussion); of
the fairly long abstract, just about half a sentence relates to where
the proposed model differs from previous work.


Recommendation:
I recommend rejection. Despite some clearly positive aspects, the two main criticisms voiced
by reviewers are serious: Weak validation and minimal
novelty from a machine learning perspective. I agree that the
neuroscience application may be interesting, but requires more validation.

If the authors want to pursue this work further, I would suggest to
perhaps consider first where to position the paper's focus.
Estimation of mutual information is
a problem that is both hard and important. Any progress here would be
welcome, and simple usefulness could offset any lack of model novelty,
but it would have to be carefully and comprehensively
evaluated. On the other hand, a focus on neuroscience applications would
require more emphasis on, and presumably more space in the paper for,
relevant experiments.